# BiPointNet: Binary Neural Network for Point Clouds

**Haotong Qin**[*1,2], **Zhongang Cai**[*3], **Mingyuan Zhang**[*3], **Yifu Ding**[1], **Haiyu Zhao**[3],
**Shuai Yi**[3], **Xianglong Liu**[†1], **Hao Su**[4]

[1]State Key Lab of Software Development Environment, Beihang University
[2]Shen Yuan Honors College, Beihang University     [3]SenseTime Research
[4]University of California, San Diego

{qinhaotong,xlliu}@nlsde.buaa.edu.cn, zjdyf@buaa.edu.cn
{caizhongang, zhangmingyuan, zhaohaiyu, yishuai}@sensetime.com
haosu@eng.ucsd.edu

## Abstract

To alleviate the resource constraint for real-time point cloud applications that run on edge devices, in this paper we present BiPointNet, the first model binarization approach for efficient deep learning on point clouds. We discover that the immense performance drop of binarized models for point clouds mainly stems from two challenges: aggregation-induced feature homogenization that leads to a degradation of information entropy, and scale distortion that hinders optimization and invalidates scale-sensitive structures. With theoretical justifications and in-depth analysis, our BiPointNet introduces *Entropy-Maximizing Aggregation* (EMA) to modulate the distribution before aggregation for the maximum information entropy, and *Layer-wise Scale Recovery* (LSR) to efficiently restore feature representation capacity. Extensive experiments show that BiPointNet outperforms existing binarization methods by convincing margins, at the level even comparable with the full precision counterpart. We highlight that our techniques are generic, guaranteeing significant improvements on various fundamental tasks and mainstream backbones. Moreover, BiPointNet gives an impressive $14.7\times$ speedup and $18.9\times$ storage saving on real-world resource-constrained devices.

## 1 Introduction

With the advent of deep neural networks that directly process raw point clouds (PointNet (Qi et al., 2017a) as the pioneering work), great success has been achieved in learning on point clouds (Qi et al., 2017b; Li et al., 2018; Wang et al., 2019a; Wu et al., 2019; Thomas et al., 2019; Liu et al., 2019b; Zhang et al., 2019b). Point cloud applications, such as autonomous driving and augmented reality, often require real-time interaction and fast response. However, computation for such applications is usually deployed on resource-constrained edge devices. To address the challenge, novel algorithms, such as Grid-GCN (Xu et al., 2020b), RandLA-Net (Hu et al., 2020), and PointVoxel (Liu et al., 2019d), have been proposed to accelerate those point cloud processing networks. While significant speedup and memory footprint reduction have been achieved, these works still rely on expensive floating-point operations, leaving room for further optimization of the performance from the model quantization perspective. Model binarization (Rastegari et al., 2016; Bulat & Tzimiropoulos, 2019; Hubara et al., 2016; Wang et al., 2020; Zhu et al., 2019; Xu et al., 2019) emerged as one of the most promising approaches to optimize neural networks for better computational and memory usage efficiency. Binary Neural Networks (BNNs) leverage 1) compact binarized parameters that take small memory space, and 2) highly efficient bitwise operations which are far less costly compared to the floating-point counterparts.

Despite that in 2D vision tasks (Krizhevsky et al., 2012; Simonyan & Zisserman, 2014; Szegedy et al., 2015; Girshick et al., 2014; Girshick, 2015; Russakovsky et al., 2015; Wang et al., 2019b;

---

[*]equal contributions
[†]corresponding author

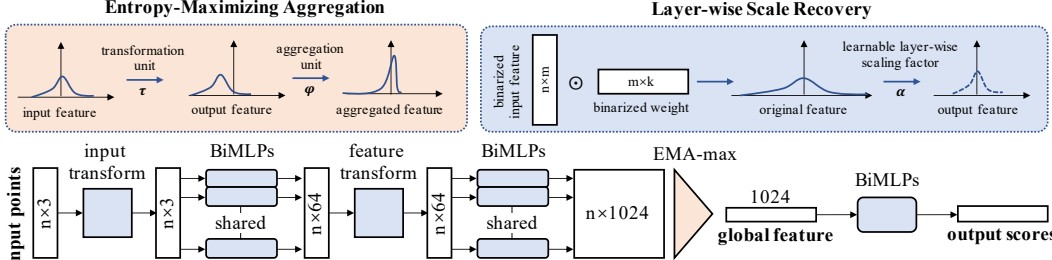

Figure 1: Overview of our BiPointNet on PointNet base model, applying Entropy-Maximizing Aggregation (EMA) and Layer-wise Scale Recovery (LSR). EMA consists of the transformation unit and the aggregation unit for maximizing the information entropy of feature after binarization. LSR with the learnable layer-wise scaling factor $\alpha$ is applied to address the scale distortion of bi-linear layers (which form the BiMLPs), flexibly restore the distorted output to reasonable values

Zhang et al., 2021) has been studied extensively by the model binarization community, the methods developed are not readily transferable for 3D point cloud networks due to the fundamental differences between 2D images and 3D point clouds. First, to gain efficiency in processing *unordered* 3D points, many point cloud learning methods rely heavily on pooling layers with large receptive field to aggregate point-wise features. As shown in PointNet (Qi et al., 2017b), global pooling provides a strong recognition capability. However, this practice poses challenges for binarization. Our analyses show that the degradation of feature diversity, a persistent problem with binarization (Liu et al., 2019a; Qin et al., 2020b; Xie et al., 2017), is significantly amplified by the global aggregation function (Figure 2), leading to homogenization of global features with limited discriminability. Second, the binarization causes immense scale distortion at the point-wise feature extraction stage, which is detrimental to model performance in two ways: the saturation of forward-propagated features and backward-propagated gradients hinders optimization, and the disruption of the scale-sensitive structures (Figure 3) results in the invalidation of their designated functionality.

In this paper, we provide theoretical formulations of the above-mentioned phenomenons and obtain insights through in-depth analysis. Such understanding allows us to propose a method that turns full-precision point cloud networks into extremely efficient yet strong binarized models (see the overview in Figure 1). To tackle the homogenization of the binarized features after passing the aggregation function, we study the correlation between the information entropy of binarization features and the performance of point cloud aggregation functions. We thus propose *Entropy-Maximizing Aggregation* (EMA) that shifts the feature distribution towards the statistical optimum, effectively improving expression capability of the global features. Moreover, given maximized information entropy, we further develop *Layer-wise Scale Recovery* (LSR) to efficiently restore the output scale that enhances optimization, which allows scale-sensitive structures to function properly. LSR uses only one learnable parameter per layer, leading to negligible storage increment and computation overhead.

Our BiPointNet is the first binarization approaches to deep learning on point clouds, and it outperforms existing binarization algorithms for 2D vision by convincing margins. It is even almost on par (within $\sim$ 1-2%) with the full-precision counterpart. Although we conduct most analysis on the PointNet baseline, we show that our methods are generic and can be readily extendable to other popular backbones, such as PointNet++ (Qi et al., 2017b), PointCNN (Li et al., 2018), DGCNN (Wang et al., 2019a), and PointConv (Wu et al., 2019), which are the representatives of mainstream categories of point cloud feature extractors. Moreover, extensive experiments on multiple fundamental tasks on the point cloud, such as classification, part segmentation, and semantic segmentation, highlight that our BiPointNet is task-agnostic. Besides, we highlight that our EMA and LSR are efficient and easy to implement in practice: in the actual test on popular edge devices, BiPointNet achieves $14.7\times$ speedup and $18.9\times$ storage savings compared to the full-precision PointNet. Our code is released at https://github.com/htqin/BiPointNet.

## 2  RELATED WORK

**Network Binarization.** Recently, various quantization methods for neural networks have emerged, such as uniform quantization (Gong et al., 2019; Zhu et al., 2020), mixed-precision quantization (Wu

et al., 2018; Yu et al., 2020), and binarization. Among these methods, binarization enjoys compact binarized parameters and highly efficient bitwise operations for extreme compression and acceleration (Rastegari et al., 2016; Qin et al., 2020a). In general, the forward and backward propagation of binarized models in the training process can be formulated as:

$$\text{Forward}: b = \texttt{sign}(x) = \begin{cases} +1, & \text{if } x \geq 0 \\ -1, & \text{otherwise} \end{cases} \qquad \text{Backward}: g_x = \begin{cases} g_b, & \text{if } x \in (-1, 1) \\ 0, & \text{otherwise} \end{cases} \quad (1)$$

where $x$ denotes the element in floating-point weights and activations, $b$ denotes the element in binarized weights $\mathbf{B_w}$ and activations $\mathbf{B_a}$. $g_x$, and $g_b$ donate the gradient $\frac{\partial C}{\partial x}$ and $\frac{\partial C}{\partial b}$, respectively, where $C$ is the cost function for the minibatch. In forward propagation, $\texttt{sign}$ function is directly applied to obtain the binary parameters. In backward propagation, the Straight-Through Estimator (STE) (Bengio et al., 2013) is used to obtain the derivative of the $\texttt{sign}$ function, avoiding getting all zero gradients. The existing binarization methods are designed to obtain accurate binarized networks by minimizing the quantization error (Rastegari et al., 2016; Zhou et al., 2016; Lin et al., 2017), improving loss function (Ding et al., 2019; Hou et al., 2017), reducing the gradient error (Liu et al., 2018; 2020), and designing novel structures and pipelines (Martinez et al., 2020). Unfortunately, we show in Sec 3 that these methods, designed for 2D vision tasks, are not readily transferable to 3D point clouds.

**Deep Learning on Point Clouds.** PointNet (Qi et al., 2017a) is the first deep learning model that processes raw point clouds directly. The basic building blocks proposed by PointNet such as MLP for point-wise feature extraction and max pooling for global aggregation (Guo et al., 2020) have become the popular design choices for various categories of newer backbones: 1) the pointwise MLP-based such as PointNet++ (Qi et al., 2017b); 2) the graph-based such as DGCNN (Xu et al., 2020b); 3) the convolution-based such as PointCNN (Li et al., 2018), PointConv (Wu et al., 2019) RS-CNN (Liu et al., 2019c) and KP-Conv (Thomas et al., 2019). Recently, methods are proposed for efficient deep learning on point clouds through novel data structuring (Xu et al., 2020b), faster sampling (Hu et al., 2020), adaptive filters (Xu et al., 2020a), efficient representation (Liu et al., 2019d) or convolution operation (Zhang et al., 2019b) . However, they still use expensive floating-point parameters and operations, which can be improved by binarization.

## 3 METHODS

Binarized models operate on efficient binary parameters, but often suffer large performance drop. Moreover, the unique characteristics of point clouds pose even more challenges. We observe there are two main problems: first, aggregation of a large number of points leads to a severe loss of feature diversity; second, binarization induces an immense scale distortion, that undermines the functionality of scale-sensitive structures. In this section, we discuss our observations, and propose our BiPointNet with theoretical justifications.

### 3.1 BINARIZATION FRAMEWORK

We first give a brief introduction to our framework that binarizes a floating-point network. For example, deep learning models on point clouds typically contain multi-layer perceptrons (MLPs) for feature extraction. In contrast, the binarized models contain binary MLPs (BiMLPs), which are composed of binarized linear (bi-linear) layers. Bi-linear layers perform the extremely efficient bitwise operations (XNOR and Bitcount) on the lightweight binary weight/activation. Specifically, the activation of the bi-linear layer is binarized to $\mathbf{B_a}$, and is computed with the binarized weight $\mathbf{B_w}$ to obtain the output $\mathbf{Z}$:

$$\mathbf{Z} = \mathbf{B_a} \odot \mathbf{B_w}, \quad (2)$$

where $\odot$ denotes the inner product for vectors with bitwise operations XNOR and Bitcount. When $B_w$ and $B_a$ denote the random variables in $\mathbf{B_w}$ and $\mathbf{B_a}$, we represent their probability mass function as $p_{B_w}(b_w)$, and $p_{B_a}(b_a)$.

Moreover, we divide the BiPointNet into units for detailed discussions. In BiPointNet, the original data or feature $\mathbf{X} \in \mathbb{R}^{n \times c}$ first enters the symmetric function $\Omega$, which represents a composite function built by stacking several permutation equivariant and permutation invariant layers (e.g., nonlinear layer, bi-linear layer, max pooling). And then, the output $\mathbf{Y} \in \mathbb{R}^{n \times k}$ is binarized to obtain

(a) Point-wise features to be aggregated.  (b) Full-precision features aggregated with max pooling.  (c) Binarized features aggregated with max pooling.  (d) Binarized features aggregated with EMA.

Figure 2: Aggregation-induced feature homogenization. (a) shows the activation of each test sample in a batch of ModelNet40. In (b)-(d), the single feature vectors pooled from all points are mapped to colors. The diversity of colors represents the diversity of pooled features. The original aggregation design is incompatible with binarization, leading to the homogenization of output features in (c), whereas our proposed EMA retains high information entropy, shown in (d)

the binary feature $\mathbf{B} \in \{-1, 1\}^{i \times k}$, where $i$ takes $n$ when the feature is modeled independently and takes 1 when the feature is aggregated globally. The single unit is thus represented as

$$\mathbf{B} = \text{sign}(\mathbf{Y}) = \text{sign}(\Omega(\mathbf{X})). \tag{3}$$

Similarly, when $B$, $Y$ and $X$ denote the random variables sampled from $\mathbf{B}$, $\mathbf{Y}$ and $\mathbf{X}$, we represent their probability mass function as $p_B(b)$, $p_Y(y)$ and $p_X(x)$.

## 3.2 ENTROPY-MAXIMIZING AGGREGATION

Unlike images pixels that are arranged in regular lattices, point clouds are sets of points without any specific order. Hence, features are usually processed in a point-wise manner and aggregated explicitly through pooling layers. Our study shows that the aggregation function is a performance bottleneck of the binarized model, due to severe homogenization as shown in Figure 2.

We apply information theory (Section 3.2.1) to quantify the effect of the loss of feature diversity, and find that global feature aggregation leads to a catastrophic loss of information entropy. In Section 3.2.2, we propose the concept of Entropy-Maximizing Aggregation (EMA) that gives the statistically maximum information entropy to effectively tackle the feature homogenization problem.

### 3.2.1 AGGREGATION-INDUCED FEATURE HOMOGENIZATION

Ideally, the binarized tensor $\mathbf{B}$ should reflect the information in the original tensor $\mathbf{Y}$ as much as possible. From the perspective of information, maximizing mutual information can maximize the information flow from the full-precision to the binarized parameters. Hence, our goal is equivalent to maximizing the mutual information $\mathcal{I}(Y; B)$ of the random variables $Y$ and $B$:

$$\arg\max_{Y,B} \mathcal{I}(Y; B) = \mathcal{H}(B) - \mathcal{H}(B \mid Y) \tag{4}$$

where $\mathcal{H}(B)$ is the information entropy, and $\mathcal{H}(B \mid Y)$ is the conditional entropy of $B$ given $Y$. $\mathcal{H}(B \mid Y) = 0$ as we use the deterministic sign function as the quantizer in binarization (see Section A.1 for details). Hence, the original objective function Eq. (4) is equivalent to:

$$\arg\max_{B} \mathcal{H}_B(B) = -\sum_{b \in \mathcal{B}} p_B(b) \log p_B(b), \tag{5}$$

where $\mathcal{B}$ is the set of possible values of $B$. We then study the information properties of max pooling, which is a common aggregation function used in popular point cloud learning models such as PointNet. Let the max pooling be the last layer $\phi$ of the multi-layer stacked $\Omega$, and the input of $\phi$ is defined as $\mathbf{X}_\phi$. The data flow of Eq. (3) can be further expressed as $\mathbf{B} = \text{sign}(\phi(\mathbf{X}_\phi))$, and the information entropy $\mathcal{H}_B$ of binarized feature $B$ can be expressed as

$$\mathcal{H}_B(X_\phi) = -\left( \sum_{x_\phi \geq 0} p_{X_\phi}(x_\phi) \right)^n \log \left( \sum_{x_\phi \geq 0} p_{X_\phi}(x_\phi) \right)^n - \left( 1 - \left( \sum_{x_\phi \geq 0} p_{X_\phi}(x_\phi) \right)^n \right) \log \left( 1 - \left( \sum_{x_\phi \geq 0} p_{X_\phi}(x_\phi) \right)^n \right) \tag{6}$$

where $n$ is the number of elements aggregated by the max pooling, and $X_\phi$ is the random variable sampled from $\mathbf{X}_\phi$. The brief derivation of Eq (6) is shown in Appendix A.2. Theorem 1 shows the information properties of max pooling with the normal distribution input on the binarized network architecture.

**Theorem 1** *For input $X_\phi$ of max pooling $\phi$ with arbitrary distribution, the information entropy of the binarized output goes to zero as $n$ goes to infinity, i.e., $\lim\limits_{n \to +\infty} \mathcal{H}_B = 0$. And there exists a constant $c$, for any $n_1$ and $n_2$, if $n_1 > n_2 > c$, we have $\mathcal{H}_{B,n_1} < \mathcal{H}_{B,n_2}$, where $n$ is the number of elements to be aggregated.*

The proof of Theorem 1 is included in Appendix A.2, which explains the severe feature homogenization after global feature pooling layers. As the number of points is typically large (e.g. 1024 points by convention in ModelNet40 classification task), it significantly reduces the information entropy $\mathcal{H}_B$ of binarized feature $\mathbf{B}$, i.e., the information of $\mathbf{Y}$ is hardly retained in $\mathbf{B}$, leading to highly similar output features regardless of the input features to pooling layer as shown in Figure 2.

Furthermore, Theorem 1 provides a theoretical justification for the poor performance of existing binarization methods, transferred from 2D vision tasks to point cloud applications. In 2D vision, the aggregation functions are often used to gather local features with a small kernel size $n$ (e.g. $n = 4$ in ResNet (He et al., 2016; Liu et al., 2018) and VGG-Net (Simonyan & Zisserman, 2014) which use $2 \times 2$ pooling kernels). Hence, the feature homogenization problem on images is not as significant as that on point clouds.

### 3.2.2 EMA for Maximum Information Entropy

Therefore, we need a class of aggregation functions that maximize the information entropy of $\mathbf{B}$ to avoid the aggregation-induced feature homogenization.

We study the correlation between the information entropy $\mathcal{H}_B$ of binary random variable $B$ and the distribution of the full-precision random variable $Y$. We notice that the sign function used in binarization has a fixed threshold and decision levels, so we get Proposition 1 about information entropy of $B$ and the distribution of $Y$.

**Proposition 1** *When the distribution of the random variable $Y$ satisfies $\sum_{y<0} p_Y(y) = \sum_{y \geq 0} p_Y(y) = 0.5$, the information entropy $\mathcal{H}_B$ is maximized.*

The proof of Proposition 1 is shown in Appendix A.3. Therefore, theoretically, there is a distribution of $Y$ that can maximize the mutual information of $\mathbf{Y}$ and $\mathbf{B}$ by maximizing the information entropy of the binary tensor $\mathbf{B}$, so as to maximally retain the information of $\mathbf{Y}$ in $\mathbf{B}$.

To maximize the information entropy $\mathcal{H}_B$, we propose the EMA for feature aggregation in BiPoint-Net. The EMA is not one, but a class of binarization-friendly aggregation layers. Modifying the aggregation function in the full-precision neural network to a EMA keeps the entropy maximized by input transformation. The definition of EMA is

$$\mathbf{Y} = \text{EMA}(\mathbf{X}_\phi) = \varphi(\tau(\mathbf{X}_\phi)), \tag{7}$$

where $\varphi$ denotes the aggregation function (e.g. max pooling and average pooling) and $\tau$ denotes the transformation unit. Note that a standard normal distribution $\mathcal{N}(0, 1)$ is assumed for $X_\phi$ because batch normalization layers are placed prior to the pooling layers by convention. $\tau$ can take many forms; we discover that a simple constant offset is already effective. The offset shifts the input so that the output distribution satisfies $\sum_{y<0} p_Y(y) = 0.5$, to maximize the information entropy of binary feature $B$. The transformation unit $\tau$ in our BiPointNet can be defined as $\tau(\mathbf{X}_\phi) = \mathbf{X}_\phi - \delta^*$.

When max pooling is applied as $\varphi$, we obtain the distribution offset $\delta^*$ for the input $X_\phi$ that maximizes the information entropy $\mathcal{H}_B$ by solving the objective function

$$\arg\max_\delta \mathcal{H}_B(\delta) = - \Big( \sum_{x_\phi \geq 0} \frac{1}{\sqrt{2\pi}} e^{-\frac{(x_\phi - \delta)^2}{2}} \Big)^n \log \Big( \sum_{x_\phi \geq 0} \frac{1}{\sqrt{2\pi}} e^{-\frac{(x_\phi - \delta)^2}{2}} \Big)^n$$

$$- \Big( 1 - \Big( \sum_{x_\phi \geq 0} \frac{1}{\sqrt{2\pi}} e^{-\frac{(x_\phi - \delta)^2}{2}} \Big)^n \Big) \log \Big( 1 - \Big( \sum_{x_\phi \geq 0} \frac{1}{\sqrt{2\pi}} e^{-\frac{(x_\phi - \delta)^2}{2}} \Big)^n \Big), \tag{8}$$

where $n$ denotes the number of elements in each batch. For each $n$, we can obtain an optimized $\delta^*_{\max}$ for Eq. (8), we include the pseudo code in the Appendix A.5.

Moreover, we derive in the Appendix A.6 that when average pooling is used as $\varphi$, the solution to its objective function is expressed as $\delta = 0$. We thus obtain $\delta^*_{\text{avg}} = 0$. This means the solution

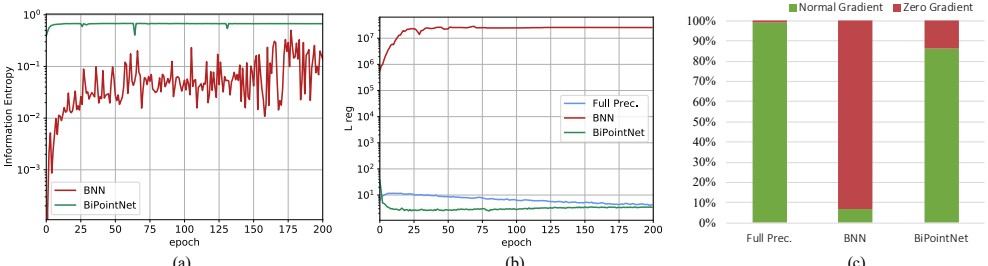

Figure 4: (a) Information entropy of the aggregated features. With EMA, our BiPointNet achieves a higher information entropy. (b) Regularizer loss comparison. Our PointNet has a low loss, indicating that the scale distortion is reduced and T-Net is not disrupted. (c) Ratio of zero-gradient activations in back-propagation. LSR alleviates the scale distortion, enhancing the optimization process

is not related to $n$. Hence, average pooling can be regarded as a flexible alternative because its performance is independent of the input number $n$.

In a nutshell, we provide two possible variants of $\varphi$: first, we show that a simple shift is sufficient to turn a max pooling layer into an EMA (EMA-max); second, average pooling can be directly used (EMA-avg) without modification as a large number of points does not undermine its information entropy, making it adaptive to the dynamically changing number of point input. Note that modifying existing aggregation functions is only one way to achieve EMA; the theory also instructs the development of new binarization-friendly aggregation functions in the future.

## 3.3 ONE-SCALE-FITS-ALL: LAYER-WISE SCALE RECOVERY

In this section, we show that binarization leads to feature scale distortion and study its cause. We conclude that the distortion is directly related to the number of feature channels. More importantly, we discuss the detriments of scale distortion from the perspectives of the functionality of scale-sensitive structures and the optimization.

To address the severe scale distortion in feature due to binarization, we propose the Layer-wise Scale Recovery (LSR). In LSR, each bi-linear layer is added only *one* learnable scaling factor to recover the original scales of *all* binarized parameters, with negligible additional computational overhead and memory usage.

### 3.3.1 SCALE DISTORTION

The scale of parameters is defined as the standard deviation $\sigma$ of their distribution. As we mentioned in Section 3.2, balanced binarized weights are used in the bi-linear layer aiming to maximize the entropy of the output after binarization, i.e., $p_{B_w}(1) = 0.5$ and $p_{B_a}(1) = 0.5$.

**Theorem 2** *When we let $p_{B_w}(1) = 0.5$ and $p_{B_a}(1) = 0.5$ in bi-linear layer to maximize the mutual information, for the binarized weight $\mathbf{B_w} \in \{-1, +1\}^{m \times k}$ and activation $\mathbf{B_a} \in \{-1, +1\}^{n \times m}$, the probability mass function for the distribution of output $\mathbf{Z}$ can be represented as $p_Z(2i - m) = 0.5^m C_m^i, i \in \{0, 1, 2, ..., m\}$. The output has approximately a normal distribution $\mathcal{N}(0, m)$.*

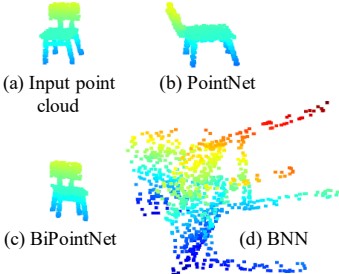

(a) Input point cloud     (b) PointNet

(c) BiPointNet     (d) BNN

Figure 3: Scale Distortion. Figures (b)-(d) show the transformed input. Compared with the input (a), the scales of (b) in full-precision PointNet and (c) in our BiPointNet are normal, while the scale of (d) in BNN is significantly distorted

The proof of Theorem 2 is found in Appendix A.4. Theorem 2 shows that given the maximized information entropy, the scale of the output features is directly related to the number of feature channels. Hence, scale distortion is pervasive as a large number of channels is the design norm of deep learning neural networks for effective feature extraction.

We discuss two major impacts of the scale distortion on the performance of binarized point cloud learning models. First, the scale distortion invalidates structures designed for 3D deep learning that

Table 1: Ablation study for our BiPointNet of various tasks on ModelNet40 (classification), ShapeNet Parts (part segmentation), and S3DIS (semantic segmentation). EMA and LSR and complementary to each other, and they are useful across all three applications

| Method | Bit-width | Aggr. | ModelNet40 | ShapeNet Parts | S3DIS | |
|---|---|---|---|---|---|---|
| | | | OA | mIoU | mIoU | OA |
| Full Prec. | 32/32 | MAX | 88.2 | 84.3 | 54.4 | 83.5 |
| | 32/32 | AVG | 86.5 | 84.0 | 51.5 | 81.5 |
| BNN | 1/1 | MAX | 7.1 | 54.0 | 9.5 | 45.0 |
| BNN-LSR | 1/1 | MAX | 4.1 | 58.7 | 2.0 | 25.4 |
| BNN-EMA | 1/1 | EMA-avg | 11.3 | 53.0 | 9.9 | 46.8 |
| | 1/1 | EMA-max | 16.2 | 47.3 | 8.5 | 47.2 |
| Ours | 1/1 | EMA-avg | 82.5 | 80.3 | 40.9 | 74.9 |
| | 1/1 | EMA-max | **86.4** | **80.6** | **44.3** | **76.7** |

are sensitive to the scale of values. For example, the T-Net in PointNet is designed to predict an orthogonal transformation matrix for canonicalization of input and intermediate features (Qi et al., 2017a). The predicted matrix is regularized by minimizing the loss term $L_{reg} = \left\| \mathbf{I} - \mathbf{Z}\mathbf{Z}^T \right\|^2$. However, this regularization is ineffective for the $\mathbf{Z}$ with huge variance as shown in Figure 3.

Second, the scale distortion leads to a saturation of forward-propagated activations and backward-propagated gradients (Ding et al., 2019). In the binary neural networks, some modules (such as sign and Hardtanh) rely on the Straight-Through Estimator (STE) Bengio et al. (2013) for feature binarization or feature balancing. When the scale of their input is amplified, the gradient is truncated instead of increased proportionally. Such saturation, as shown in Fig 4(c), hinders learning and even leads to divergence.

### 3.3.2 LSR FOR OUTPUT SCALE RECOVERY

To recover the scale and adjustment ability of output, we propose the LSR for bi-linear layers in our BiPointNet. We design a learnable layer-wise scaling factor $\alpha$ in our LSR. $\alpha$ is initialized by the ratio of the standard deviations between the output of bi-linear and full-precision counterpart:

$$\alpha_0 = \sigma(\mathbf{A} \otimes \mathbf{W})/\sigma(\mathbf{B_a} \odot \mathbf{B_w}), \qquad (9)$$

where $\sigma$ denotes as the standard deviation. And the $\alpha$ is learnable during the training process. The calculation and derivative process of the bi-linear layer with our LSR are as follows:

$$\text{Forward} : \mathbf{Z} = \alpha(\mathbf{B_a} \odot \mathbf{B_w}) \qquad \text{Backward} : g_\alpha = g_\mathbf{Z}(\mathbf{B_a} \odot \mathbf{B_w}), \qquad (10)$$

where $g_\alpha$ and $g_\mathbf{Z}$ denotes the gradient $\frac{\partial C}{\partial \alpha}$ and $\frac{\partial C}{\partial \mathbf{Z}}$, respectively. By applying the LSR in BiPointNet, we mitigate the scale distortion of output caused by binarization.

Compared to existing methods, the advantages of LSR is summarized in two folds. First, LSR is efficient. It not only abandons the adjustment of input activations to avoid expensive inference time computation, but also recovers the scale of all weights parameters in a layer collectively instead of expensive restoration in a channel-wise manner (Rastegari et al., 2016). Second, LSR serves the purpose of scale recovery that we show is more effective than other adaptation such as minimizing quantization errors (Qin et al., 2020b; Liu et al., 2018).

## 4 EXPERIMENTS

In this section, we conduct extensive experiments to validate the effectiveness of our proposed BiPointNet for efficient learning on point clouds. We first ablate our method and demonstrate the contributions of EMA and LSR on three most fundamental tasks: classification on ModelNet40 (Wu et al., 2015), part segmentation on ShapeNet (Chang et al., 2015), and semantic segmentation on S3DIS (Armeni et al., 2016). Moreover, we compare BiPointNet with existing binarization methods where our designs stand out. Besides, BiPointNet is put to the test on real-world devices with limited computational power and achieve extremely high speedup ($14.7\times$) and storage saving ($18.9\times$). The details of the datasets and the implementations are included in the Appendix E.

Table 2: Comparison of binarization methods on PointNet. EMA is critical; even if all methods are equipped with our EMA, our LSR outperforms others with least number of scaling factors. OA: Overall Accuracy

| Method | Bit-width | Aggr. | # Factors | OA |
|---|---|---|---|---|
| Full Prec. | 32/32 | MAX | - | 88.2 |
|  | 32/32 | AVG | - | 86.5 |
| BNN | 1/1 | MAX | 0 | 7.1 |
|  | 1/1 | EMA-avg | 0 | 11.3 |
|  | 1/1 | EMA-max | 0 | 16.2 |
| IR-Net | 1/1 | MAX | 10097 | 7.3 |
|  | 1/1 | EMA-avg | 10097 | 22.0 |
|  | 1/1 | EMA-max | 10097 | 63.5 |
| Bi-Real | 1/1 | MAX | 10097 | 4.0 |
|  | 1/1 | EMA-avg | 10097 | 77.0 |
|  | 1/1 | EMA-max | 10097 | 77.5 |
| ABC-Net | 1/1 | MAX | 51 | 4.1 |
|  | 1/1 | EMA-avg | 51 | 68.9 |
|  | 1/1 | EMA-max | 51 | 77.8 |
| XNOR++ | 1/1 | MAX | 18 | 4.1 |
|  | 1/1 | EMA-avg | 18 | 73.8 |
|  | 1/1 | EMA-max | 18 | 78.4 |
| XNOR | 1/1 | MAX | 28529 | 64.9 |
|  | 1/1 | EMA-avg | 28529 | 78.2 |
|  | 1/1 | EMA-max | 28529 | 81.9 |
| Ours | 1/1 | MAX | 18 | 4.1 |
|  | 1/1 | EMA-avg | 18 | 82.5 |
|  | 1/1 | EMA-max | 18 | **86.4** |

Table 3: Our methods on mainstream backbones. We use XNOR as a strong baseline for comparison. The techniques in our BiPointNet are generic to point cloud learning. Hence, they are easily extendable to other backbones

| Base Model | Method | Bit-width | Aggr. | OA |
|---|---|---|---|---|
| PointNet (Vanilla) | Full Prec. | 32/32 | MAX | 86.8 |
|  | XNOR | 1/1 | MAX | 61.0 |
|  | Ours | 1/1 | EMA-max | 85.6 |
| PointNet | Full Prec. | 32/32 | MAX | 88.2 |
|  | XNOR | 1/1 | MAX | 64.9 |
|  | Ours | 1/1 | EMA-max | 86.4 |
| PointNet++ | Full Prec. | 32/32 | MAX | 90.0 |
|  | XNOR | 1/1 | MAX | 63.1 |
|  | Ours | 1/1 | EMA-max | 87.8 |
| PointCNN | Full Prec. | 32/32 | AVG | 90.0 |
|  | XNOR | 1/1 | AVG | 83.0 |
|  | Ours | 1/1 | EMA-avg | 83.8 |
| DGCNN | Full Prec. | 32/32 | MAX | 89.2 |
|  | XNOR | 1/1 | MAX | 51.5 |
|  | Ours | 1/1 | EMA-max | 83.4 |
| PointConv | Full Prec. | 32/32 | – | 90.8 |
|  | XNOR | 1/1 | – | 83.1 |
|  | Ours | 1/1 | – | 87.9 |

## 4.1 ABLATION STUDY

As shown in Table 1, the binarization model baseline suffers a catastrophic performance drop in the classification task. EMA and LSR improve performance considerably when used alone, and they further close the gap between the binarized model and the full-precision counterpart when used together.

In Figure 4, we further validate the effectiveness of EMA and LSR. We show that BiPointNet with EMA has its information entropy maximized during training, whereas the vanilla binarized network with max pooling gives limited and highly fluctuating results. Also, we make use of the regularization loss $L_{reg} = \left\| \mathbf{I} - \mathbf{Z}\mathbf{Z}^T \right\|_F$ for the feature transformation matrix of T-Net in PointNet as an indicator, the $L_{reg}$ of the BiPointNet with LSR is much smaller than the vanilla binarized network, demonstrating LSR's ability to reduce the scale distortion caused by binarization, allowing proper prediction of orthogonal transformation matrices.

Moreover, we also include the results of two challenging tasks, part segmentation, and semantic segmentation, in Table 1. As we follow the original PointNet design for segmentation, which concatenates pointwise features with max pooled global feature, segmentation suffers from the information loss caused by the aggregation function. EMA and LSR are proven to be effective: BiPointNet is approaching the full precision counterpart with only $\sim 4\%$ mIoU difference on part segmentation and $\sim 10.4\%$ mIoU gap on semantic segmentation. The full results of segmentation are presented in Appendix E.6.

## 4.2 COMPARATIVE EXPERIMENTS

In Table 2, we show that our BiPointNet outperforms other binarization methods such as BNN (Hubara et al., 2016), XNOR (Rastegari et al., 2016), Bi-Real (Liu et al., 2018), ABC-Net (Lin

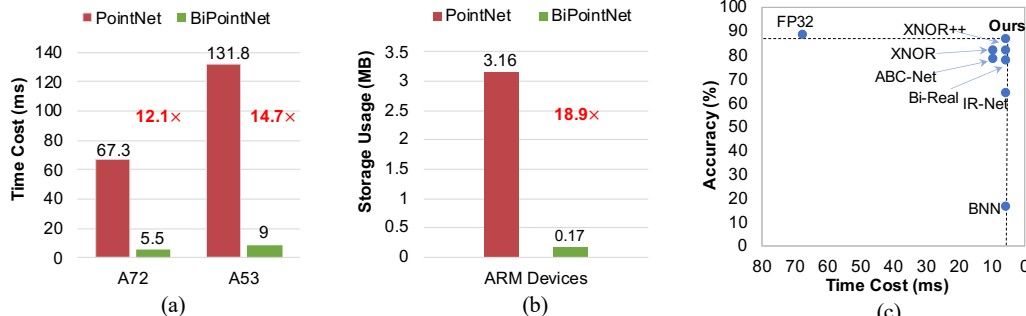

Figure 5: (a) Time cost comparison. Our BiPointNet achieves $14.7\times$ speedup on ARM A72 CPU device. (b) Storage usage comparison. Our BiPointNet enjoys $18.9\times$ storage saving on all devices. (c) Speed vs accuracy trade-off plot. We evaluate various binarization methods (with our EMA-max) upon PointNet architecture on ARM A72 CPU device, our BiPointNet is the leading method in both speed and accuracy

et al., 2017), XNOR++ (Bulat & Tzimiropoulos, 2019), and IR-Net (Qin et al., 2020b). Although these methods have been proven effective in 2D vision, they are not readily transferable to point clouds due to aggregation-induced feature homogenization.

Even if we equip these methods with our EMA to mitigate information loss, our BiPointNet still performs better. We argue that existing approaches, albeit having many scaling factors, focus on minimizing quantization errors instead of recovering feature scales, which is critical to effective learning on point clouds. Hence, BiPointNet stands out with a negligible increase of parameters that are designed to restore feature scales. The detailed analysis of the performance of XNOR is found in Appendix C. Moreover, we highlight that our EMA and LSR are generic, and Table 3 shows improvements across several mainstream categories of point cloud deep learning models, including PointNet (Qi et al., 2017a), PointNet++ (Qi et al., 2017b), PointCNN (Li et al., 2018), DGCNN (Wang et al., 2019a), and PointConv (Wu et al., 2019).

## 4.3 Deployment Efficiency on Real-world Devices

To further validate the efficiency of BiPointNet when deployed into the real-world edge devices, we further implement our BiPointNet on Raspberry Pi 4B with 1.5 GHz 64-bit quad-core ARM CPU Cortex-A72 and Raspberry Pi 3B with 1.2 GHz 64-bit quad-core ARM CPU Cortex-A53.

We compare our BiPointNet with the PointNet in Figure 5(a) and Figure 5(b). We highlight that BiPointNet achieves $14.7\times$ inference speed increase and $18.9\times$ storage reduction over PointNet, which is recognized as a fast and lightweight model itself. Moreover, we implement various binarization methods over PointNet architecture and report their real speed performance on ARM A72 CPU device. As Figure 5(c), our BiPointNet surpasses all existing binarization methods in both speed and accuracy. Note that all binarization methods adopt our EMA and report their best accuracy, which is the important premise that they can be reasonably applied to binarize the PointNet.

## 5 Conclusion

We propose BiPointNet, the first binarization approach for efficient learning on point clouds. We build a theoretical foundation to study the impact of binarization on point cloud learning models, and proposed EMA and LSR in BiPointNet to improve the performance. BiPointNet outperforms existing binarization methods, and it is easily extendable to a wide range of tasks and backbones, giving an impressive $14.7\times$ speedup and $18.9\times$ storage saving on resource-constrained devices. Our work demonstrates the great potential of binarization. We hope our work can provide directions for future research.

**Acknowledgement** This work was supported by National Natural Science Foundation of China (62022009, 61872021), Beijing Nova Program of Science and Technology (Z191100001119050), and State Key Lab of Software Development Environment (SKLSDE-2020ZX-06).

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

# APPENDIX FOR BIPOINTNET

## A  MAIN PROOFS AND DISCUSSION

### A.1  PROOF OF ZERO CONDITIONAL ENTROPY

In our BiPointNet, we hope that the binarized tensor $\mathbf{B}$ reflects the information in the original tensor $\mathbf{Y}$ as much as possible. From the perspective of information, our goal is equivalent to maximizing the mutual information $\mathcal{I}(Y; B)$ of the random variables $Y$ and $B$:

$$\underset{Y,B}{\arg\max}\ \mathcal{I}(Y; B) \tag{11}$$

$$= \sum_{y\in\mathcal{Y}, b\in\mathcal{B}} p_{(Y,B)}(y,b) \log \frac{p_{(Y,B)}(y,b)}{p_Y(y)p_B(b)} \tag{12}$$

$$= \sum_{y\in\mathcal{Y}, b\in\mathcal{B}} p_{(Y,B)}(y,b) \log \frac{p_{(Y,B)}(y,b)}{p_Y(y)} - \sum_{y\in\mathcal{Y}, b\in\mathcal{B}} p_{(Y,B)}(y,b) \log p_B(b) \tag{13}$$

$$= \sum_{y\in\mathcal{Y}, b\in\mathcal{B}} p_Y(y) p_{B|Y=y}(b) \log p_{B|Y=y}(b) - \sum_{y\in\mathcal{Y}, b\in\mathcal{B}} p_{(Y,B)}(y,b) \log p_B(b) \tag{14}$$

$$= \sum_{y\in\mathcal{Y}} p_Y(y)\Big(\sum_{b\in\mathcal{B}} p_{B|Y=y}(b) \log p_{B|Y=y}(b)\Big) - \sum_{b\in\mathcal{B}}\Big(\sum_{y} p_{(Y,B)}(y,b)\Big) \log p_B(b) \tag{15}$$

$$= -\sum_{y\in\mathcal{Y}} p(y)\mathcal{H}(B \mid Y=y) - \sum_{b\in\mathcal{B}} p_B(b) \log p_B(b) \tag{16}$$

$$= -\mathcal{H}(B \mid Y) + \mathcal{H}(B) \tag{17}$$

$$= \mathcal{H}(B) - \mathcal{H}(B \mid Y), \tag{18}$$

where $p_{(Y,B)}$ and $p_Y$, $p_B$ are the joint and marginal probability mass functions of these discrete variables. $\mathcal{H}(B)$ is the information entropy, and $\mathcal{H}(B|Y)$ is the conditional entropy of $B$ given $Y$. According to the Eq. (15) and Eq. (18), the conditional entropy $\mathcal{H}(Y \mid X)$ can be expressed as

$$\mathcal{H}(B \mid Y) = \sum_{y\in\mathcal{Y}} p_Y(y)\Big(\sum_{b\in\mathcal{B}} p_{B|Y=y}(b) \log p_{B|Y=y}(b)\Big). \tag{19}$$

Since we use the deterministic sign function as the quantizer in binarization, the value of $B$ fully depends on the value of $Y$, $p_{B|Y=y}(b) = 0$ or $1$ in Eq. (4), i.e., every value $y$ has a fixed mapping to a binary value $b$. Then we have

$$\mathcal{H}(B \mid Y) = \sum_{y\in\mathcal{Y}} p_Y(y)(0 + 0 + \cdots + 0) = 0. \tag{20}$$

Hence, the original objective function is equivalent to maximizing the information entropy $\mathcal{H}(B)$:

$$\underset{B}{\arg\max}\ \mathcal{H}_B(B) = -\sum_{b\in\mathcal{B}} p_B(b) \log p_B(b). \tag{21}$$

### A.2  PROOFS OF THEOREM 1

**Theorem 1** *For input $X_\phi$ of max pooling $\phi$ with arbitrary distribution, the information entropy of the binarized output to zero as $n$ to infinity, i.e., $\lim_{n\to+\infty} \mathcal{H}_B = 0$. And there is a constant c, for any $n_1$ and $n_2$, if $n_1 > n_2 > c$, we have $\mathcal{H}_{B,n_1} < \mathcal{H}_{B,n_2}$, where $n$ is the number of aggregated elements.*

**Proof.** We obtain the correlation between the probability mass function of input $\mathbf{X}_\phi$ and output $\mathbf{Y}$ of max pooling, intuitively, all values are negative to give a negative maximum value:

$$\sum_{y<0} p_Y(y) = \Big( \sum_{x_\phi<0} p_{X_\phi}(x_\phi) \Big)^n. \tag{22}$$

Since the sign function is applied as the quantizer, the $\mathcal{H}_B(B)$ of binarized feature can be expressed as Eq. (6).

(1) When $X_\phi$ obeys a arbitrary distribution, the probability mass function $p_{X_\phi}(x_\phi)$ must satisfies $\sum_{x_\phi<0} p_{X_\phi}(x_\phi) \leq 1$. According to Eq. (6), let $t = \sum_{x_\phi<0} p_{X_\phi}(x_\phi)$, we have

$$\lim_{n\to\infty} \mathcal{H}_B(X_\phi) = \lim_{n\to\infty} -t^n \ \log \ t^n - (1-t)^n \ \log \ (1-t)^n \tag{23}$$

$$= - \Big( \lim_{n\to\infty} t^n \Big) \ \log \ \Big( \lim_{n\to\infty} t^n \Big) - \Big( \lim_{n\to\infty} (1-t)^n \Big) \ \log \ \Big( \lim_{n\to\infty} (1-t)^n \Big) \tag{24}$$

$$= - 0 \ \log \ 0 - 1 \ \log \ 1 \tag{25}$$

$$= 0 \tag{26}$$

(2) For any $n \geq 1$, we can obtain the representation of the information entropy $\mathcal{H}_{B,n}(X_\phi)$:

$$\begin{aligned}
\mathcal{H}_{B,n}(X_\phi) = &- \Big( \sum_{x_\phi<0} p_{X_\phi}(x_\phi) \Big)^n \log \Big( \sum_{x_\phi<0} p_{X_\phi}(x_\phi) \Big)^n \\
&- \Big( 1 - \Big( \sum_{x_\phi<0} p_{X_\phi}(x_\phi) \Big)^n \Big) \log \Big( 1 - \Big( \sum_{x_\phi<0} p_{X_\phi}(x_\phi) \Big)^n \Big),
\end{aligned} \tag{27}$$

Let $p_n = \Big( \sum_{x_\phi<0} p_{X_\phi}(x_\phi) \Big)^n$, the $\mathcal{H}_{B,n}(p_n)$ can be expressed as

$$\mathcal{H}_{B,n}(p_n) = -p_n \log p_n - (1-p_n) \log(1-p_n), \tag{28}$$

and the derivative of $\mathcal{H}_{B,n}(p_n)$ is

$$\frac{d\,\mathcal{H}_{B,n}(p_n)}{d\,p_B(p_n)} = \log \Big( \frac{1-p_n}{p_n} \Big), \tag{29}$$

the $\mathcal{H}_{B,n}(p_n)$ is maximized when $p_n$ takes 0.5, and is positive correlation with $p_n$ when $p_n < 0.5$ since the $\frac{d\,\mathcal{H}_{B,n}(p_n)}{d\,p_B(p_n)} > 0$ when $p_n < 0.5$.

Therefore, when the constant $c$ satisfies $p_c = \Big( \sum_{x_\phi<0} p_{X_\phi}(x_\phi) \Big)^c \geq 0.5$, given the $n_1 > n_2 > c$, we have $p_{n_1} < p_{n_2} < p_c$, and $\mathcal{H}_{B,n_1}(X_\phi) < \mathcal{H}_{B,n_2}(X_\phi) < \mathcal{H}_{B,c}(X_\phi)$.

$\square$

### A.3 PROOFS OF PROPOSITION 1

**Proposition 1** *When the distribution of the random variable $Y$ satisfies $\sum_{y<0} p_Y(y) = \sum_{y\geq0} p_Y(y) = 0.5$, the information entropy $\mathcal{H}_B$ is maximized.*

**Proof.** According to Eq (5), we have

$$\mathcal{H}_B(B) = - \sum_{b\in\mathcal{B}} p_B(b) \log p_B(b) \tag{30}$$

$$= - p_B(-1) \log p_B(-1) - p_B(1) \log p_B(1) \tag{31}$$

$$= - p_B(-1) \log p_B(-1) - (1 - p_B(-1) \log (1 - p_B(-1))) \,. \tag{32}$$

Then we can get the derivative of $\mathcal{H}_B(B)$ with respect to $p_B(-1)$

$$\frac{d\,\mathcal{H}_B(B)}{d\,p_B(-1)} = -\left(\log p_B(-1) + \frac{p_B(-1)}{p_B(-1)\ln 2}\right) + \left(\log\left(1 - p_B(-1)\right) + \frac{1 - p_B(-1)}{(1 - p_B(-1))\ln 2}\right) \tag{33}$$

$$= -\log p_B(-1) + \log\left(1 - p_B(-1)\right) - \frac{1}{\ln 2} + \frac{1}{\ln 2} \tag{34}$$

$$= \log\left(\frac{1 - p_B(-1)}{p_B(-1)}\right). \tag{35}$$

When we let $\frac{d\,\mathcal{H}_B(B)}{d\,p_B(-1)} = 0$ to maximize the $\mathcal{H}_B(B)$, we have $p_B(-1) = 0.5$. Sine the deterministic $sign$ function with the zero threshold is applied as the quantizer, the probability mass function of $B$ is represented as

$$p_B(b) = \begin{cases} \sum\limits_{y<0} p_Y(y)\,dy, & \text{if } b = -1 \\ \sum\limits_{y\geq 0} p_Y(y)\,dy, & \text{if } b = 1, \end{cases} \tag{36}$$

and when the information entropy is maximized, we have

$$\sum_{y<0} p_Y(y)\,dy = 0.5. \tag{37}$$

$\square$

### A.4 DISCUSSION AND PROOFS OF THEOREM 2

The bi-linear layers are widely used in our BiPointNet to model each point independently, and each linear layer outputs an intermediate feature. The calculation of the bi-linear layer is represented as Eq. (2). Since the random variable $B$ is sampled from $\mathbf{B_w}$ or $\mathbf{B_a}$ obeying Bernoulli distribution, the probability mass function of $B$ can be represented as

$$p_B(b) = \begin{cases} p, & \text{if } b = +1 \\ 1 - p, & \text{if } b = -1, \end{cases} \tag{38}$$

where $p$ is the probability of taking the value $+1$. The distribution of output $\mathbf{Z}$ can be represented by the probability mass function of $\mathbf{B_w}$ and $\mathbf{B_a}$.

**Proposition 2** *In bi-linear layer, for the binarized weight $\mathbf{B_w} \in \{-1, +1\}^{m\times k}$ and activation $\mathbf{B_a} \in \{-1, +1\}^{n\times m}$ with probability mass function $p_{B_w}(1) = p_w$ and $p_{B_a}(1) = p_a$, the probability mass function for the distribution of output $\mathbf{Z}$ can be represented as $p_Z(2i - m) = C_m^i(1 - p_w - p_a + 2p_w p_a)^i(p_w + p_a - 2p_w p_a)^{m-i}, i \in \{0, 1, 2, ..., m\}$.*

**Proof.** To simplify the notation in the following statements, we define $\mathbf{A} = \mathbf{B_a}$ and $\mathbf{W} = \mathbf{B_w}$. Then, for each element $\mathbf{Z}_{i,j}$ in output $\mathbf{Z} \in \{-1, +1\}^{n\times k}$, we have

$$x_{i,j} = \sum_{k=1}^{m} \mathbf{A}_{i,k} \times \mathbf{W}_{k,j}. \tag{39}$$

Observe that $\mathbf{A}_{i,k}$ is independent to $\mathbf{W}_{k,j}$ and the value of both variables are either $-1$ or $+1$. Therefore, the discrete probability distribution of $\mathbf{A}_{i,k} \times \mathbf{W}_{k,j}$ can be defined as

$$p(x) = \begin{cases} p_w p_a + (1 - p_w) \times (1 - p_a), & \text{if } x = 1 \\ p_w \times (1 - p_a) + (1 - p_w) \times p_a, & \text{if } x = -1 \\ 0, & \text{otherwise.} \end{cases} \tag{40}$$

Simplify the above equation

$$p(x) = \begin{cases} 1 - p_w - p_a + 2p_w p_a, & \text{if } x = 1 \\ p_w + p_a - 2p_w p_a, & \text{if } x = -1 \\ 0, & \text{otherwise.} \end{cases} \qquad (41)$$

Notice that $x_{i,j}$ can be parameterized as a binomial distribution. Then we have

$$\Pr(x_{i,j} = l - (m - l)) = C_m^l (1 - p_w - p_a + 2p_w p_a)^l (p_w + p_a - 2p_w p_a)^{m-l}. \qquad (42)$$

Observe that $p_Z$ obeys the same distribution as $x_{i,j}$. Finally, we have

$$p_Z(2i - m) = C_m^i (1 - p_w - p_a + 2p_w p_a)^i (p_w + p_a - 2p_w p_a)^{m-i}, i \in \{0, 1, 2, ..., m\}. \qquad (43)$$

$\square$

Proposition 2 shows that the output distribution of the bi-linear layer depends on the probability mass functions of binarized weight and activation. Then we present the proofs of Theorem 2.

**Theorem 2** *When we let $p_{B_w}(1) = 0.5$ and $p_{B_a}(1) = 0.5$ in bi-linear layer to maximize the mutual information, for the binarized weight $\mathbf{B_w} \in \{-1, +1\}^{m \times k}$ and activation $\mathbf{B_a} \in \{-1, +1\}^{n \times m}$, the probability mass function for the distribution of output $\mathbf{Z}$ can be represented as $p_Z(2i - m) = 0.5^m C_m^i, i \in \{0, 1, 2, ..., m\}$. The distribution of output is approximate normal distribution $\mathcal{N}(0, m)$.*

**Proof.** First, we prove that the distribution of $Z$ can be approximated as a normal distribution. For bi-linear layers in our BiPointNet, all weights and activations are binarized, which can be represented as $\mathbf{B_w}$ and $\mathbf{B_a}$, respectively. And the value of an element $z_{(i,j)}$ in $\mathbf{Z}$ can be expressed as

$$z_{(i,j)} = \sum_{k=1}^{m} \left( b_{w(i,k)} \times b_{a(k,j)} \right),$$

and the value of the element $b_{w(i,k)} \times b_{a(k,j)}$ can be expressed as

$$b_{w(i,k)} \times b_{a(k,j)} = \begin{cases} 1, & \text{if } b_{w(i,k)} \veebar b_{a(k,j)} = 1 \\ -1, & \text{if } b_{w(i,k)} \veebar b_{a(k,j)} = -1. \end{cases} \qquad (44)$$

The $b_{w(i,k)} \times b_{a(k,j)}$ only can take from two values and its value can be considered as the result of one Bernoulli trial. Thus for the random variable $Z$ sampled from the output tensor $\mathbf{Z}$, the probability mass function, $p_Z$ can be expressed as

$$p_Z(2i - m) = C_m^i p_e^k (1 - p_e)^{n-k}, \qquad (45)$$

where $p_e$ denotes the probability that the element $b_{w(i,k)} \times b_{a(k,j)}$ takes 1. Note that the Eq. (45) is completely equivalent to the representation in Proposition 2. According to the *De Moivre–Laplace* theorem, the normal distribution $\mathcal{N}(\mu, \sigma^2)$ can be used as an approximation of the binomial distribution under certain conditions, and the $p_Z(2i - m)$ can be approximated as

$$p_Z(2i - m) = C_m^i p_e^k (1 - p_e)^{n-k} \simeq \frac{1}{\sqrt{2\pi n p_e (1 - p_e)}} e^{-\frac{(k - n p_e)^2}{2 n p_e (1 - p_e)}}, \qquad (46)$$

and then, we can get the mean $\mu = 0$ and variance $\sigma = \sqrt{m}$ of the approximated distribution $\mathcal{N}$ with the help of equivalent representation of $p_Z$ in Proposition 2. Now we give proof of this below.

According to Proposition 2, when $p_w = p_a = 0.5$, we can rewrite the equation as

$$p_Z(2i - m) = 0.5^m C_m^i, i \in \{0, 1, 2, ..., m\}. \tag{47}$$

Then we move to calculate the mean and standard variation of this distribution. The mean of this distribution is defined as

$$\mu(p_Z) = \sum (2i - m) 0.5^m C_m^i, i \in \{0, 1, 2, ..., m\}. \tag{48}$$

By the virtue of binomial coefficient, we have

$$(2i - m)0.5^m C_m^i + (2(m - i) - m)0.5^m C_m^{m-i} = 0.5^m((2i - m)C_m^i + (m - 2i)C_m^{m-i}) \tag{49}$$
$$= 0.5^m((2i - m)C_m^i + (m - 2i)C_m^i) \tag{50}$$
$$= 0. \tag{51}$$

Besides, when $m$ is an even number, we have $(2i - m)0.5^m C_m^i = 0, i = \frac{m}{2}$. These equations prove the symmetry of function $(2i - m)0.5^m C_m^i$. Finally, we have

$$\mu(p_Z) = \sum (2i - m)0.5^m C_m^i, i \in \{0, 1, 2, ..., m\} \tag{52}$$
$$= \sum ((2i - m)0.5^m C_m^i + (2(m - i) - m)0.5^m C_m^{m-i}), i \in \{0, 1, 2, ..., \frac{m}{2}\} \tag{53}$$
$$= 0. \tag{54}$$

The standard variation of $p_Z$ is defined as

$$\sigma(p_Z) = \sqrt{\left(\sum |2i - m|^2 0.5^m C_m^i\right)} \tag{55}$$
$$= \sqrt{\sum (4i^2 - 4im + m^2) 0.5^m C_m^i} \tag{56}$$
$$= \sqrt{0.5^m \left(4 \sum i^2 C_m^i - 4m \sum i C_m^i + m^2 \sum C_m^i\right)}. \tag{57}$$

To calculate the standard variation of $p_Z$, we use Binomial Theorem and have several identical equations:

$$\sum C_m^i = (1 + 1)^m = 2^m \tag{58}$$
$$\sum i C_m^i = m(1 + 1)^{m-1} = m2^{m-1} \tag{59}$$
$$\sum i^2 C_m^i = m(m + 1)(1 + 1)^{m-2} = m(m + 1)m2^{m-2}. \tag{60}$$

These identical equations help simplify Eq. (57):

$$\sigma(p_Z) = \sqrt{0.5^m \left(4 \sum i^2 C_m^i - 4m \sum i C_m^i + m^2 \sum C_m^i\right)} \tag{61}$$
$$= \sqrt{0.5^m (4m(m + 1)2^{m-2} - 4m^2 2^{m-1} + m^2 2^m)} \tag{62}$$
$$= \sqrt{0.5^m ((m^2 + m)2^m - 2m^2 2^m + m^2 2^m)} \tag{63}$$
$$= \sqrt{0.5^m (m2^m)} \tag{64}$$
$$= \sqrt{m}. \tag{65}$$

Now we proved that, the distribution of output is approximate normal distribution $\mathcal{N}(0, m)$. $\square$

## A.5 Discussion of the Optimal $\delta$ for EMA-max

When the $X_\phi \sim \mathcal{N}(0, 1)$, the objective function of EMA-max to obtain optimal $\delta^*$ is represented as Eq. (8). It is difficult to directly solve the objective function. To circumvent this issue, we use Monte Carlo simulation to approximate the value of the optimal $\delta_{\max}^*$ as shown Algorithm 1.

---

**Algorithm 1** Monte Carlo Simulation for EMA-max

   **Input:** The number $n$ of points to be aggregated; the number of simulations m (e.g. 10000)
   **Output:** Estimated optimal $\delta_{\max}^*$ for EMA-max
1: Creating an empty list $F$ (represents elements sampled form distribution of aggregated feature)
2: **for** $i = 0$ to m **do**
3:    Creating an empty list $T_i$ (representing one channel of input feature)
4:    **for** $j = 0$ to $n$ **do**
5:       Sampling an element $e_{ij}$ from the distribution $\mathcal{N}(0, 1)$
6:       Adding the sampled element $e_{ij}$ to the list $T_i$
7:    **end for**
8:    Adding an element represents the aggregated feature $\text{MAX}(T_i)$ to $F$
9: **end for**
10: Estimating the optimal $\delta_{\max}^*$ as $\delta_{\max}^* = \text{Median}(F)$ (follow Proposition 1)

---

## A.6 Discussion of the Optimal $\delta$ for EMA-avg

When the $X_\phi \sim \mathcal{N}(\delta, 1)$, the $Y \sim \mathcal{N}(\delta, n^{-1})$ and the objective function of EMA-avg for obtaining optimal $\delta_{\text{avg}}^*$ can be represented as

$$
\begin{aligned}
\arg\max_\delta \mathcal{H}_B(\delta) = &- \Big( \sum_{x_\phi < 0} \frac{1}{n^{-1}\sqrt{2\pi}} \, e^{-\frac{(x_\phi - \delta)^2}{2}} \Big) \log \Big( \sum_{x_\phi < 0} \frac{1}{n^{-1}\sqrt{2\pi}} \, e^{-\frac{(x_\phi - \delta)^2}{2}} \Big) \\
&- \Big( \sum_{x_\phi \geq 0} \frac{1}{n^{-1}\sqrt{2\pi}} \, e^{-\frac{(x_\phi - \delta)^2}{2}} \Big) \log \Big( \sum_{x_\phi \geq 0} \frac{1}{n^{-1}\sqrt{2\pi}} \, e^{-\frac{(x_\phi - \delta)^2}{2}} \Big).
\end{aligned}
\tag{66}
$$

The solution of Eq. (66) is expressed as $\delta = 0$, we thus obtain $\delta_{\text{avg}}^* = 0$. This means the solution is not related to $n$.

# B Implementation of BiPointNet on ARM Devices

## B.1 Overview

We further implement our BiPointNet on Raspberry Pi 4B with 1.5 GHz 64-bit quad-core ARM Cortex-A72 and Raspberry Pi 3B with 1.2 GHz 64-bit quad-core ARM Cortex-A53, and test the real speed that one can obtain in practice. Although the PointNet is a recognized high-efficiency model, the inference speed of BiPointNet is much faster. Compared to PointNet, BiPointNet enjoys up to $14.7\times$ speedup and $18.9\times$ storage saving.

We utilize the SIMD instruction SSHL on ARM NEON to make inference framework daBNN (Zhang et al., 2019a) compatible with our BiPointNet and further optimize the implementation for more efficient inference.

## B.2 Implementation Details

Figure 6 shows the detailed structures of six PointNet implementations. In Full-Precision version (a), BN is merged into the later fully connected layer for speedup, which is widely chosen for deployment in real-world applications. In Binarization version (b)(c)(d)(e), we have to keep BN unmerged due to the binarization of later layers. Instead, we merge the scaling factor of LSR into BN layers. The `HardTanh` function is removed because it does not affect the binarized value of

input for the later layers. We test the quantization for the first layer and last layer in the variants (b)(c)(d)(e). In the last variant(f), we drop the BN layers during training. The scaling factor is ignored during deployment because it does not change the sign of the output.

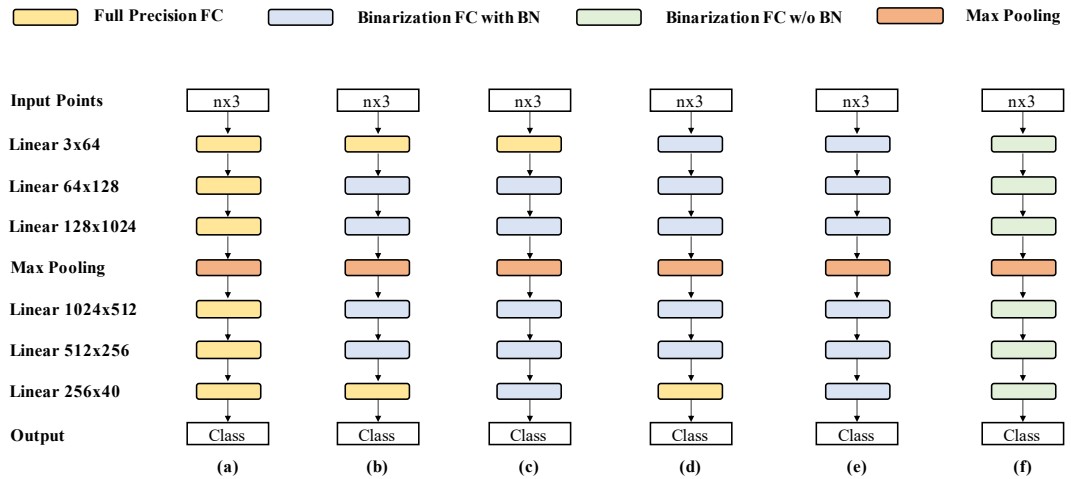

Figure 6: Structures of different PointNet implementations. Three fully connected layers are used in all six variants: Full Precision FC, Binarization FC with BN, Binarization FC w/o BN. Full Precision FC contains a full precision fully connected layer and a ReLU layer. Original BN is merged into the later layer. Binarization FC with BN also contains two layers: a quantized fully connected layer and a batch normalization layer. Binarization FC w/o BN is formed by a single quantized fully connected layer

## B.3 ABLATION ANALYSIS OF TIME COST AND QUANTIZATION SENSITIVITY

| Setup | Bit-width | FL | LL | BN | OA | Storage & Saving Ratio | Time & Speedup Ratio | |
|-------|-----------|-------|-------|------------|-------|------------------------|----------------------|----------------------|
| | | | | | | | A72 | A53 |
| (a) | 32/32 | 32/32 | 32/32 | Merged | 86.8 | 3.16MB / 1.0× | 131ms / 1.0× | 67ms / 1.0× |
| (b) | 1/1 | 32/32 | 32/32 | Not Merged | 85.62 | 0.17MB / 18.9× | 9.0ms / 14.7× | 5.5ms / 12.1× |
| (c) | 1/1 | 32/32 | 1/1 | Not Merged | 84.60 | 0.12MB / 26.3× | 9.0ms / 14.7× | 5.3ms / 12.6× |
| (d) | 1/1 | 1/1 | 32/32 | Not Merged | 5.31 | 0.16MB / 19.7× | 11.5ms / 11.4× | 6.5ms / 10.3× |
| (e) | 1/1 | 1/1 | 1/1 | Not Merged | 4.86 | 0.12MB / 26.3× | 11.4ms / 11.5× | 6.4ms / 10.4× |
| (f) | 1/1 | 32/32 | 32/32 | Not Used | 85.13 | 0.15MB / 21.0× | 8.1ms / 16.1× | 4.8ms / 13.9× |

Table 4: Comparison of different configurations in deployment on ARM devices. The storage-saving ratio and speedup ratio are calculated according to the full precision model as the first row illustrates. All the models use PointNet as the base model and EMA-max as the aggregation function. The accuracy performance is reported on the point cloud classification task with the ModelNet40 dataset. FL: First Layer; LL: Last Layer

Table 4 shows the detailed configuration including overall accuracy, storage usage, and time cost of the above-mentioned six implementations. The result shows that binarization of the middle fully connected layers can extremely speed up the original model. We achieve 18.9× storage saving, 14.7× speedup on A72, and 12.1× speed on A53. The quantization of the last layer further helps save storage consumption and improves the speed with a slight performance drop. However, the quantization of the first layer causes a drastic drop in accuracy without discernible computational cost reduction. The variant (f) without BN achieves comparable performance with variant (b). It suggests that our LSR method could be an ideal alternative to the original normalization layers to achieve a fully quantized model except for the first layer.

## C    COMPARISON BETWEEN LAYER-WISE SCALE RECOVERY AND OTHER METHODS

In this section, we will analyze the difference between the LSR method with other model binarization methods. Theorem 2 shows the significance of recovering scale in point cloud learning. However, IRNet and BiReal only consider the scale of weight but ignore the scale of input features. Therefore, these two methods cannot recover the scale of output due to scale distortion on the input feature. A major difference between these two methods is that LSR opts for layer-wise scaling factor while XNOR opts for point-wise one. Point-wise scale recovery needs dynamical computation during inference while our proposed LSR only has a layer-wise global scaling factor, which is independent of the input. As a result, our method can achieve higher speed in practice.

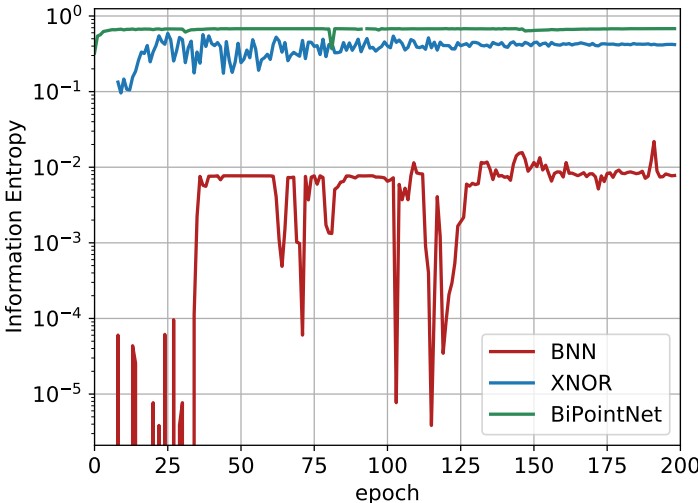

Figure 7: The information entropy of BNN, XNOR and our BiPointNet

Table 3 shows that XNOR can alleviate the aggregation-induced feature homogenization. The point-wise scaling factor helps the model to achieve comparable adjustment capacity as full-precision linear layers. Therefore, although XNOR suffers from feature homogenization at the beginning of the training process, it can alleviate this problem with the progress of training and achieve acceptable performance, as shown in Figure 7.

## D    COMPARISON WITH OTHER EFFICIENT LEARNING METHODS

We compare our computation speedup and storage savings with several recently proposed methods to accelerate deep learning models on point clouds. Note that the comparison is for reference only; tests are conducted on different hardware, and for different tasks. Hence, direct comparison *cannot* give any meaningful conclusion. In Table 5, we show that BiPointNet achieves the most impressive acceleration.

## E    EXPERIMENTS

### E.1    DATASETS

**ModelNet40:** ModelNet40 (Wu et al., 2015) for part segmentation. The ModelNet40 dataset is the most frequently used dataset for shape classification. ModelNet is a popular benchmark for point cloud classification. It contains 12,311 CAD models from 40 representative classes of objects.

Table 5: Comparison between BiPointNet and other approaches to efficient learning on point clouds. Grid-GCN (Xu et al., 2020b) leverages novel data structuring strategy; RAND-LA Net (Hu et al., 2020) designs a faster sampling method; PointVoxel (Liu et al., 2019d) proposes an efficient representation. These works, albeit achieving high performance, are not as effective as our binarization method in terms of model acceleration. The asterisk indicates the vanilla version

| Method | Hardware | Dataset | Base Model | Metric/ Performance | Speedup |
|---|---|---|---|---|---|
| BiPointNet | ARM Cortex-A72 | ModelNet4 | PointNet* | OA/85.6 | 12.1× |
| BiPointNet | ARM Cortex-A53 | ModelNet40 | PointNet* | OA/85.6 | 14.7× |
| Grid-GCN | RTX 2080 GPU | S3DIS | PointNet | mIoU/53.2 | 1.62× |
| RandLA-Net | RTX 2080Ti GPU | S3DIS | PointNet* | mIoU/70.0 | 1.04× |
| PointVoxel | GTX 1080Ti GPU | ShapeNet | PointNet | mIoU/46.9 | 2.46× |

**ShapeNet Parts:** ShapeNet Parts (Chang et al., 2015) for part segmentation. ShapeNet contains 16,881 shapes from 16 categories, 2,048 points are sampled from each training shape. Each shape is split into two to five parts depending on the category, making up to 50 parts in total.

**S3DIS:** S3DIS for semantic segmentation (Armeni et al., 2016). S3DIS includes 3D scan point clouds for 6 indoor areas including 272 rooms in total, each point belongs to one of 13 semantic categories. We follow the official code (Qi et al., 2017a) for training and testing.

### E.2    IMPLEMENTATION DETAILS OF BIPOINTNET

We follow the popular PyTorch implementation of PointNet and the recent geometric deep learning codebase (Fey & Lenssen, 2019) for the implementation of PointNet baselines. Our BiPointNet is built by binarizing the full-precision PointNet. All linear layers in PointNet except the first and last one are binarized to bi-linear layer, and we select Hardtanh as our activation function instead of ReLU when we binarize the activation before the bi-linear layer. For the part segmentation task, we follow the convention (Wu et al., 2014; Yi et al., 2016) to train a model for each of the 16 classes. We also provide our PointNet baseline under this setting.

Following previous works, we train 200 epochs, 250 epochs, 128 epochs on point cloud classification, part segmentation, semantic segmentation respectively. To stably train the binarized models, we use a learning rate of 0.001 with Adam and Cosine Annealing learning rate decay for all binarized models on all three tasks.

### E.3    MORE BACKBONES

We also propose four other models: BiPointCNN, BiPointNet++, BiDGCCN, and BiPointConv, which are binarized versions of PointCNN (Li et al., 2018), PointNet++ (Qi et al., 2017b), DGCNN (Wang et al., 2019a), and PointConv (Wu et al., 2019), respectively. This is attributed to the fact that all these variants have characteristics in common, such as linear layers for point-wise feature extraction and global pooling layers for feature aggregation (except PointConv, which does not have explicit aggregators). In PointNet++, DGCNN, and PointConv, we keep the first layer and the last layer full-precision and binarize all the other layers. In PointCNN, we keep every first layer of XConv full precision and keep the last layer of the classifier full precision.

### E.4    BINARIZATION METHODS

For comparison, we implement various representative binarization methods for 2D vision, including BNN (Hubara et al., 2016), XNOR-Net (Rastegari et al., 2016), Bi-Real Net (Liu et al., 2018), XNOR++ (Bulat & Tzimiropoulos, 2019), ABC-Net (Lin et al., 2017), and IR-Net (Qin et al., 2020b), to be applied on 3D point clouds. Note that the Case 1 version of XNOR++ is used in our experiments for a fair comparison, which applies layerwise learnable scaling factors to minimize the quantization error. These methods are implemented according to their open-source code or the description in their papers, and we take reference of their 3x3 convolution design when implementing the corresponding bi-linear layers. We follow their training process and hyperparameter

settings, but note that the specific shortcut structure in Bi-Real and IR-Net is ignored since it only applies to the ResNet architecture.

### E.5 TRAINING DETAILS

Our BiPointNet is trained from scratch (random initialization) without leveraging any pre-trained model. Amongst the experiments, we apply Adam as our optimizer and use the cosine annealing learning rate scheduler to stably optimize the networks. To evaluate our BiPointNet on various network architectures, we mostly follow the hyper-parameter settings of the original papers (Qi et al., 2017a; Li et al., 2018; Qi et al., 2017b; Wang et al., 2019a).

### E.6 DETAILED RESULTS OF SEGMENTATION

We present the detailed results of part segmentation on ShapeNet Part in Table 6 and semantic segmentation on S3DIS in Table 7. The detailed results further prove the conclusion of Section 4.1 as EMA and LSR improve performance considerably in most of the categories (instead of huge performance in only a few categories). This validates the effectiveness and robustness of our method.

Table 6: Detailed results of our BiPointNet for part segmentation on ShapeNet Parts.

| | aggr. | mean | aero | bag | cap | car | chair | ear phone | guitar | knife | lamp | laptop | motor | mug | pistol | rocket | skate board | table |
|---|---|---|---|---|---|---|---|---|---|---|---|---|---|---|---|---|---|---|
| # shapes | | 2690 | 76 | 55 | 898 | 3758 | 69 | 787 | 392 | 1547 | 451 | 202 | 184 | 283 | 66 | 152 | 5271 | |
| FP | max | 84.3 | 83.6 | 79.4 | 92.5 | 76.8 | 90.8 | 70.2 | 91.0 | 85.6 | 81.9 | 95.6 | 64.4 | 93.5 | 80.9 | 54.5 | 70.6 | 81.5 |
| FP | avg | 84.0 | 83.4 | 78.5 | 90.8 | 76.3 | 90.0 | 73.1 | 90.8 | 84.3 | 80.8 | 95.5 | 61.7 | 93.8 | 81.6 | 56.2 | 72.2 | 81.8 |
| BNN | max | 54.0 | 35.1 | 48.1 | 65.5 | 26.5 | 55.8 | 57.1 | 48.8 | 62.2 | 48.6 | 90.1 | 23.1 | 68.3 | 57.5 | 31.3 | 43.7 | 66.8 |
| BNN | ema-avg | 53.0 | 39.8 | 46.5 | 57.5 | 24.1 | 58.2 | 56.2 | 44.0 | 50.0 | 53.0 | 81.0 | 16.9 | 48.8 | 36.3 | 25.7 | 43.7 | 63.3 |
| BNN | ema-max | 47.3 | 37.9 | 46.2 | 44.6 | 24.1 | 61.3 | 38.2 | 33.5 | 42.6 | 50.8 | 48.6 | 16.9 | 49.0 | 25.2 | 26.8 | 43.7 | 50.30 |
| LSR | max | 58.7 | 41.5 | 46.2 | 80.2 | 39.2 | 75.3 | 46.0 | 47.8 | 75.5 | 50.0 | 93.8 | 25.4 | 51.0 | 60.2 | 36.2 | 43.7 | 61.4 |
| Ours | ema-avg | 80.3 | 79.3 | 71.9 | 85.5 | 66.1 | 87.7 | 65.6 | 84.1 | 82.8 | 76.0 | 94.8 | 42.7 | 91.8 | 75.9 | 47.2 | 59.1 | 79.7 |
| Ours | ema-max | 80.6 | 79.5 | 69.7 | 86.1 | 67.4 | 88.6 | 68.5 | 87.4 | 83.0 | 74.9 | 95.1 | 44.8 | 91.6 | 76.3 | 47.7 | 56.9 | 79.5 |

Table 7: Detailed results of our BiPointNet for semantic segmentation on S3DIS.

| method | aggr | overall mIoU | overall acc. | area1 (mIoU/acc.) | area2 (mIoU/acc.) | area3 (mIoU/acc.) | area4 (mIoU/acc.) | area5 (mIoU/acc.) | area6 (mIoU/acc.) | ceiling IoU | floor IoU | wall IoU | beam IoU | column IoU | window IoU | door IoU | table IoU | chair IoU | sofa IoU | bookcase IoU | board IoU | clutter IoU |
|---|---|---|---|---|---|---|---|---|---|---|---|---|---|---|---|---|---|---|---|---|---|---|
| FP | max | 54.4 | 83.5 | 61.7/86.2 | 38.0/76.8 | 62.4/88.0 | 45.0/82.4 | 45.3/83.3 | 70.0/89.2 | 91.1 | 93.8 | 72.8 | 50.3 | 34.6 | 52.0 | 58.0 | 55.8 | 51.3 | 14.5 | 44.4 | 43.4 | 45.2 |
| FP | avg | 51.5 | 81.5 | 59.9/84.6 | 35.4/72.4 | 61.2/87.2 | 43.8/81.2 | 42.0/81.2 | 68.2/88.3 | 90.1 | 89.1 | 71.7 | 46.1 | 33.7 | 53.5 | 53.8 | 53.8 | 47.8 | 9.4 | 40.4 | 38.7 | 41.8 |
| BNN | max | 9.5 | 45.0 | 9.6/44.0 | 9.8/50.5 | 8.3/41.9 | 9.3/42.5 | 9.5/45.8 | 9.8/41.6 | 45.5 | 40.6 | 28.1 | 0 | 0 | 0 | 0 | 0 | 7.7 | 0 | 0 | 0 | 2.1 |
| BNN | ema-avg | 9.9 | 46.8 | 7.6/36.6 | 11.2/51.2 | 7.1/36.5 | 9.8/46.0 | 11.4/54.8 | 8.6/41.6 | 51.5 | 35.1 | 32.1 | 0 | 0 | 0 | 0.6 | 9.3 | 0 | 0 | 0 | 0 | 0.6 |
| BNN | ema-max | 8.5 | 47.2 | 7.7/44.0 | 10.1/54.4 | 7.1/46.8 | 7.8/39.7 | 7.6/49.2 | 7.2/45.3 | 50.8 | 43.5 | 15.9 | 0 | 0 | 0 | 0 | 0 | 0 | 0 | 0 | 0 | 0 |
| LSR | max | 2.0 | 25.4 | 2.0/26.0 | 2.1/27.0 | 2.0/25.7 | 1.8/22.8 | 2.0/25.8 | 1.9/24.5 | 25.4 | 0 | 0 | 0 | 0 | 0 | 0 | 0 | 0 | 0 | 0 | 0 | 0 |
| Ours | ema-avg | 40.9 | 74.9 | 47.1/75.8 | 29.1/68.3 | 48.0/79.9 | 34.2/73.2 | 34.7/76.1 | 53.3/79.8 | 84.6 | 84.6 | 60.5 | 32.0 | 19.0 | 39.6 | 43.0 | 43.5 | 39.2 | 5.8 | 30.5 | 18.5 | 31.3 |
| Ours | ema-max | **44.3** | **76.7** | **50.9/78.3** | **31.0/70.3** | **53.4/82.4** | **36.6/73.9** | **36.9/77.6** | **57.9/82.3** | **85.1** | **86.1** | **62.6** | **34.5** | **23.8** | **43.0** | **48.0** | **45.7** | **40.6** | **9.6** | **36.9** | **26.2** | **33.9** |

