# OpenReview forum: "BiPointNet: Binary Neural Network for Point Clouds"
_ICLR.cc/2021/Conference — ICLR 2021 Poster_

### Official Review · AnonReviewer2 · 2020-10-28
**A novel work on the binarization of point cloud models for time efficiency and storage saving.**

**Rating:** 7
**Confidence:** 3

**Review:**

The paper proposes a binarization approach for efficient deep learning on point clouds, called BiPointNet. The authors claim that the immense performance drop of binarized models is caused by the aggregation-induced feature homogenization and scale distortion. The authors propose Entropy-Maximizing Aggregation(EMA) and Layer-wise Scale Recovery(LSR) to reduce the side-effects of binarization. Experiment results demonstrate that the proposed BIPointNet is able to achieve state-of-the-art results and gives an impressive speedup and storage saving.

Besides the major contribution of the paper, the writing of the paper is concise and the illustrations are clear. However, the paper mainly focuses on PointNet-kind of structure, such as PointNet++, and DGCNN, etc. It would be better if the authors could give more discussion on the application of the EMA and LSR on more advanced methods, such as KPConv and PointConv, etc.

---

> ### Author Response · Authors · 2020-11-15
> **Response to AnonReviewer #2**
>
> We would like to first express our deep appreciation for your insightful comments. Our response to your suggestion can be found below:
>
> **Q1**: Give more discussion on the application of the EMA and LSR on more advanced methods.
>
> **A1**: Our paper evaluates the proposed method on three representative network architectures (PointNet for Pointwise MLP Networks, PointCNN for Convolution-based Networks, DGCNN for Graph-based Networks [1]); We agree that PointConv and KPConv are very important work, the computationally intensive operations in these architectures can be significantly accelerated by the binarization module with our LSR (such as the MLP layers in PointConv), and the EMA can be applied to aggregators (such as the max-pooling in KPConv) to avoid feature homogenization. We further add more discussion in the revised version of our paper. In fact, we have implemented our BiPointNet on PointConv: our method achieves good results, outperforming XNOR-Net by 4.8% with much fewer parameters. We are in the process of implementing our methods on more architectures and try our best to add these results in the revised version.
>
> |Base Model&nbsp;&nbsp;&nbsp;&nbsp;|Method&nbsp;&nbsp;&nbsp;&nbsp;|Bit-width&nbsp;&nbsp;&nbsp;&nbsp;|OA|
> |----------------------|------------------------| -----------|-----------|
> | PointConv| FP32| 32/32| 90.8%  |
> |		        | XNOR-Net&nbsp;&nbsp;&nbsp;&nbsp;&nbsp;&nbsp;| 1/1	  | 83.1%	|
> | 		        | Ours   	              	| 1/1	  |  87.9%	|
>
> [1] Guo et. al., Deep Learning for 3D Point Clouds: A Survey, TPAMI 2020

---

### Official Review · AnonReviewer3 · 2020-10-28
**Strong empirical and theoretical results**

**Rating:** 7
**Confidence:** 3

**Review:**

This paper proposes a method for learning binary neural networks on point cloud inputs. They provide an entropy analysis of the binarized distributions as well as an offset transform to maximize entropy. Then they analyze the scale of binary activations and propose a learnable scaling to reduce the effects of scale distortion. They show that their method is competitive with other binary neural network methods and even full precision methods. Finally they show performant speed and storage results on a Raspberry Pi.

Strengths:
- Strong empirical results backed by theoretical analysis
- Experiments are comprehensive and show competitive results on accuracy and speed

Concerns:
- If speed vs accuracy is the main trade-off, I would like to see a more thorough evaluation of all the models and baselines on a speed vs accuracy tradeoff plot

Given the strong empirical and theoretical results, I would recommend an accept. I would still like to see the authors strengthen their paper with a more detailed speed/accuracy trade off.

---

> ### Author Response · Authors · 2020-11-15
> **Response to AnonReviewer #3**
>
> We would like to first express our deep appreciation for your insightful comments. Our response to your suggestion can be found below:
>
> **Q1**:  More thorough evaluation of all the models and baselines on a speed vs accuracy tradeoff plot.
>
> **A1**: We agree that a speed vs accuracy tradeoff plot is helpful. We thus implement and evaluate more quantization methods based on PointNet architecture on ARM devices, and complete a speed vs accuracy trade-off scatter plot. Since we cannot add this plot in our response, we present our accuracy vs speed results on ARM devices in the table below:
>
> | CPU  &nbsp;&nbsp;| Method &nbsp;&nbsp;&nbsp;&nbsp;| Acc.(%)&nbsp;&nbsp;&nbsp;&nbsp; | Time cost (ms) |
> |-----------|------------------|----------|----------------------|
> | A72 	| FP32        	| 88.2   	| 67.3     |
> |  	| BNN         	| 16.2   	| 5.5    	|
> |  	| IR-Net       	| 63.5   	| 5.5    	|
> |  	| Bi-Real Net&nbsp;&nbsp;&nbsp;&nbsp;| 77.5   	| 5.5    	|
> |  	| ABC-Net 	| 77.8   	| 9.2    	|
> |  	| XNOR-Net 	| 81.9   	| 9.7    	|
> |  	| Ours         	| 86.4   	| 5.5    	|
> | A53 	| FP32        	| 88.2   	| 131.8    |
> |  	| BNN         	| 16.2   	| 9.0    	|
> |  	| IR-Net       	| 63.5   	| 9.0    	|
> |  	| Bi-Real Net	| 77.5   	| 9.0    	|
> |  	| ABC-Net 	| 77.8   	| 15.6      |
> |  	| XNOR-Net 	| 81.9   	| 15.7      |
> |  	| Ours         	| 86.4   	| 9.0    	|
>
> The results show that our BiPointNet outperforms others in both speed and accuracy, and the speed of our BiPointNet is much faster than the FP32 model with only a small drop of accuracy. And the speed vs accuracy trade-off plot will be added in the revised version. Note that through optimization of implementation on ARM devices, the models with fixed scaling factor (IR-Net, Bi-Real Net, and ours) can be as fast as the BNN baseline with no scaling factors. However, models that use dynamically computed scaling factors (XNOR-Net and ABC-Net) suffer extra computational burdens.
>
> Besides, more complicated base models, such as PointCNN, DGCNN and PointConv, contain operations that are difficult to implement and optimize on ARM devices (such as KNN and FPS), especially within the short response time. Nevertheless, we are in the progress of implementing our method on more architectures for 3D point clouds and make them easy to deploy on resource-limited devices.

---

### Official Review · AnonReviewer4 · 2020-10-29
**Review for ICLR 2021 submission "BiPointNet"**

**Rating:** 8
**Confidence:** 3

**Review:**

This paper proposes a method for binarization of neural networks of 3d point clouds. Two modules of entropy maximum aggregation and layer-wise scale recovery are proposed to conquer the problems of discrimination loss induced by feature homogenization and scale imbalance, which are caused by model binarization. The authors provide theoretical analysis about the proposed method. Experiments on various backbones and tasks demonstrate the effectiveness of the proposed method. A practical implantation of BiPointNet on ARM also demonstrates significant speedups over PointNet and large memory savings.

Strength:
1. Binarization of CNN models designed for 2D images has been studied in the past years, this paper extends this problem into 3D point cloud models. The authors show that the existing methods for binarization of 2D CNN models can not work well on this new problem.
2. For this new problem, the authors analysis its performance degradation based on PointNet and proposed effective solutions.
3. Experiments on several tasks show that the proposed method can obtain highly compact models with acceptable accuracy degradation. Experiments on other backbones also show that the proposed method is general, although its analysis is based on PointNet.

For the weakness, I only have some minor comments.
1. The discussion on related work could be enlarged. For example, the following papers are well known point cloud networks proposed recently
(a) PointConv: Deep Convolutional Networks on 3D Point Clouds. CVPR 2019
(b) Relation-Shape Convolutional Neural Network for Point Cloud Analysis. CVPR 2019
(c) ShellNet: Efficient Point Cloud Convolutional Neural Networks using Concentric Shells Statistics. ICCV 2019

Mixed precision quantization is also an active direction after binarization of neural networks, it could also be mentioned as a possible improvement in the future.
(d) Mixed Precision Quantization of Convnets via Differentiable Neural Architecture Search. ICLR 2019
(e) Search What You Want: Barrier Panelty NAS for Mixed Precision Quantization. ECCV 2020.

2. Some references miss publication type, i.e.,  conference or journal and where they are published.
3. Figure 1 can be improved. It is unclear how LSR works in the whole framework.

---

> ### Author Response · Authors · 2020-11-15
> **Response to AnonReviewer #4**
>
> We are deeply grateful for the reviewer’s support of our work and we thank the reviewer for the constructive and helpful suggestions. We provide additional discussions below:
>
> **Q1**: The discussion on related work could be enlarged.
>
> **A1**: We will add more discussions on the latest works on point clouds, including those mentioned by the reviewer. Our paper evaluates the proposed method on three representative network architectures (PointNet for Pointwise MLP Networks, PointCNN for Convolution-based Networks, DGCNN for Graph-based Networks [1]); we agree that PointConv, RS-CNN, and ShellNet are also important works to be discussed in the revised version of our paper. Moreover, despite the limited response time, we have implemented various binarization methods based on PointConv, and our method also achieves outstanding results: ours outperform XNOR-Net by 4.8% with much fewer parameters. We are working on implementing our method on more base models.
>
> |Base Model&nbsp;&nbsp;&nbsp;&nbsp;|Method&nbsp;&nbsp;&nbsp;&nbsp;|Bit-width&nbsp;&nbsp;&nbsp;&nbsp;|OA|
> |----------------------|------------------------| -----------|-----------|
> | PointConv| FP32| 32/32| 90.8%  |
> |		        | XNOR-Net&nbsp;&nbsp;&nbsp;&nbsp;&nbsp;&nbsp;| 1/1	  | 83.1%	|
> | 		        | Ours   	              	| 1/1	  |  87.9%	|
>
> We will also add more discussion on mixed precision quantization in the final version. Mixed precision quantization is also a popular approach to network compression and acceleration. Unlike binarization that pursues extreme compression and acceleration, mixed-precision quantization achieves a balance between speed and accuracy by adjusting the quantization accuracy of different layers. We are also interested to explore mixed precision quantization for point cloud model acceleration in our future work.
>
> [1] Guo et. al., Deep Learning for 3D Point Clouds: A Survey, TPAMI 2020
>
> **Q2**: Some references miss publication type.
>
> **A2**: We will carefully correct our references in the revised version.
>
>
> **Q3**: Figure 1 can be improved.
>
> **A3**: We will improve Figure 1 and add related explanations in the revised version. The proposed LSR is applied to the bi-linear layers (which form the BiMLPs in Figure 1) in our BiPointNet, the learnable layerwise scaling factors are applied to recover scales of the features obtained by XNOR-Bitcount operation ($\mathbf Z = \alpha(\mathbf{B_a}\odot \mathbf{B_w})$).

---

### Official Review · AnonReviewer1 · 2020-10-29
**a good paper but probably not good enough**

**Rating:** 4
**Confidence:** 5

**Review:**

This paper proposes a method to apply binary networks on point clouds. To my knowledge this is the first time that this is attempted so unquestionably the topic of the paper is interesting. From what the authors show a vanilla BNN (XNOR-Net) applied to point clouds does not give very good results and for this reason the authors identify solutions that boil down to applying a shift and a scaling. This is really my main problem with the paper: the proposed methods are too simple and the accompanying theory does not look to be so convincing in order to theoretically support the contributions which are just a simple shift and scaling. Actually the authors show that one of the variants of their method  can be reduced to average pooling which does not require some sophisticated explanation to convince the reader why it works.

Moreover the proposed learnable layer-wise scaling factor is not new and was previously usedat a) layer-level ( Towards Accurate Binary Convolutional Neural Network, Lin et al, NeurIPS’17) and b) channel-level (XNOR-Net++: Improved Binary Neural Networks, Bulat&Tzimiropoulos, BMVC’19). In fact, the problem itself is known since at least 2016, where in the (XNOR-Net: ImageNet Classification Using Binary Convolutional Neural Networks, Rastegari etal, ECCV’16) identifies this problem and proposes an analytically-computed scaling factor.

Other issues:
“even global pooling provides strong recognition performance. However, this practice poses challenges for binarization” – there is no justification provided of why pooling may pose issues for binarization. Avg and max-pooling is used with success in contemporary binary networks.

Existing BNNs “are not readily transferable to point clouds.” – In the paper it is mentioned that this is shown and evaluated in the method section. However, many of the listed methods are not in fact evaluated. This is especially important for methods that also learn to recover the scaling factors.

Given that for image classification, at least for ResNet the last layer before the linear classifier is a global pooling operation, how does the proposed EMA changes the results when applied to image classification, on a ResNet18 on Imagenet? Are there any improvements measurable in that case too?

“Despite that model binarization has been studied extensively in 2D vision tasks (Krizhevsky et al.,2012; Simonyan & Zisserman, 2014; Szegedy et al., 2015; Girshick et al., 2014; Girshick, 2015;Russakovsky et al., 2015; Wang et al., 2019b)" – the cited works don’t support the author statement, since none them perform binarization.

$\textbf{Final Rating}$

Based on the authors' responses during the rebuttal period, I don't believe that the paper makes a sufficient contribution for ICLR. Hence I will stick to my original score.

---

> ### Author Response · Authors · 2020-11-15
> **Response to AnonReviewer #1 (1/3)**
>
> We thank the reviewer for the feedback and comments. We respond to the concerns below:
>
> **Q1**: The proposed methods are too simple and the accompanying theory is not convincing. One of the variants can be reduced to average pooling.**
>
> **A1**: Regarding “too simple”: We regard the simplicity of our approach as a “merit” rather than a disadvantage, which makes our approach easily deployable in practice. Reviewer 2, 3, and 4 have provided positive feedback on the significance and plausibility of our method. In fact, to pursue “extreme compression and acceleration”, simplicity is critical for the performance of the binarized model. To this end, we carefully design our techniques to be efficient and lightweight: mean shifting and scaling lead to minimum computational overhead and additional storage. We also highlight that the LSR uses only one parameter per layer, further reduces model storage and enhances fast computation.
>
> Regarding our theory: PointNet is an architecture that is fundamentally different from Convolutional Neural Networks used in 2D images. We have shown in the paper that 2D binarization methods are NOT readily transferable to 3D (Sec 3). Therefore, a new theory is needed to instruct the binarization of PointNet.
>
> Our EMA theory plays this role. Instructed by our theory, we have invented EMA-max, which outperforms existing baselines and one of our own variants, EMA-avg (average pooling), as shown in Table 1 and 2. This is aligned with the fact that max pooling outperforms average pooling in the full precision PointNet.
>
> For LSR, we prove in Theorem 2 that the number of feature channels causes a scale distortion for 3D. Through detailed evaluation, we identify two detriments of the scale distortion: first, scale sensitive structures are invalidated (Figure 4 indicates without scale recovery, the transformed point cloud has a large scaling error); second, optimization is hindered (Figure 3c shows without scale recovery, the majority of the gradient is truncated). Therefore, we not only provide complete theoretical proofs, but supplement them with empirical evidence and in-depth analyses.
>
>
> **Q2**: The proposed learnable layer-wise scaling factor is not new and was previously used in existing works.
>
> **A2**: The 3D point clouds exhibit a fundamentally different data structure compared to 2D images. Hence, the problems associated with model binarization in these two domains are not the same: we identify that the *scale distortion* imposes a prominent impact on binarized PointNet (Section 3.3) whereas existing methods aim to address the *quantization error* problem in CNNs used for 2D vision tasks. We design LSR to directly tackle the scale distortion whereas existing binarization methods for CNNs are less effective due to the misalignment of the optimization target and the problem.
>
> Specifically, the layerwise scaling factor in our LSR is initialized with the ratio of standard deviation statistics between the output features of FP32 and binary networks ($\alpha_0 = \frac{\sigma(\mathbf A\otimes \mathbf W) }{ \sigma(\mathbf{B_a}\odot \mathbf{B_w})}$), which aims to recover the layerwise scale of the output features of binarized layers to that of FP32 layers. In contrast, the optimization target of XNOR-Net is minimizing the absolute (quantization) error ($\mathop{\arg\min}\limits_{\alpha,\textbf{B}}||\mathbf{W}-\alpha \mathbf{B}||^{2}$), which aims to find the optimal approximation of FP32 output features. Moreover, recent methods with scaling factors such as Bi-Real Net and IR-Net, just use one set of factors to approximate the weights only for efficient inference. They do not provide means to adapt to the scale distortion of the input activations.
>
> In addition to XNOR-Net, IR-Net, Bi-Real Net that we have already evaluated in Table 2, we have followed the reviewer’s recommendation to include ABC-Net and XNOR-Net++, which apply analytically-computed layerwise scaling factor and learnable scaling factor respectively, in our experiments (see results below). The results are supportive of our theory. First, XNOR-Net outperforms more recent methods such as Bi-Real Net and IR-Net, but these recent methods perform better than XNOR-Net in 2D vision tasks. This discrepancy highlights that there are different challenges of model binarization in CNNs for 2D vision and PointNet for 3D vision. Second, despite that XNOR-Net minimizes absolute quantization error of both the input activations and weight parameters, LSR is able to achieve the best accuracy using only one parameter per layer, demonstrating that LSR is more effective than existing methods in tackling the scale distortion problem that is critical for 3D point clouds. Note that the binarized models without our EMA do not converge in the training process (results are as low as 4.1%), which shows that the EMA is necessary for binarized PointNet.

---

> > ### Author Response · Authors · 2020-11-15
> > **Response to AnonReviewer #1 (2/3)**
> >
> > (A2 Continued)
> >
> > | Method  | Bit-width&nbsp;&nbsp; &nbsp;&nbsp;| Aggr.    | Acc.  	|
> > |-----------------------|---------------|-------------------|----------|
> > | ABC-Net            | 1/1	  | MAX            	 | 4.1%    |
> > | ABC-Net       	 | 1/1	  | EMA-avg     	| 68.9%	|
> > | ABC-Net       	 | 1/1	  | EMA-max  &nbsp;&nbsp;     	| 77.8%	|
> > | XNOR-Net++  &nbsp;&nbsp;| 1/1	  | MAX            	| 4.1%     |
> > | XNOR-Net++	 | 1/1	  | EMA-avg     	| 73.8%	|
> > | XNOR-Net++	 | 1/1	  | EMA-max    	| 78.4%	|
> > | Ours              	 | 1/1	  | MAX            	| 4.1%     |
> > | Ours              	 | 1/1	  | EMA-avg     	| 82.5%	|
> > | Ours   	              	| 1/1	  | EMA-max    	| 86.4%	|
> >
> > **Q3**: There is no justification of why pooling may pose issues for binarization. Avg and max-pooling is used with success in contemporary binary networks.
> >
> > **A3**: We were as surprised as you to see that global pooling would cause significant issues in binarizing PointNet. But that is what our experiments show and we have to respect the experiment results, which is indeed rooted in the difference between PointNet and CNNs. We have discussed in Sec 3.2 that global aggregation function (global pooling) is a major challenge for binarizing PointNet. We also highlight that this challenge is more prominent in PointNet than in CNNs in Sec 3.2.1, which explains why pooling causes fewer problems for contemporary binarized neural networks. We reiterate and elaborate these two points below:
> >
> > (1) We qualitatively (Figure 2) and quantitatively (Table 1 and 2) show that max pooling, a very common global aggregation function, leads to feature homogenization. To explain and address this problem, we utilize information theory to develop EMA, a class of point cloud aggregation functions that are binarization-friendly.
> >
> > (2) Theorem 1 shows that the severity of feature homogenization is associated with the number of elements participating in the pooling. In PointNet, the global max pooling aggregates 1024 points whereas in 2D vision, the kernels used in the binarized layers of popular backbones such as ResNet or VGG are small (typically 2x2). Hence, fewer elements in 2D vision leads to less feature homogenization. Please refer to A5 for the global pooling layer in ResNet.
> >
> >
> > **Q4**: Existing binarization methods are not evaluated on the point clouds, especially those methods that learn to recover the scaling factors.
> >
> > **A4**: We have shown in the paper that existing binarization works are designed for 2D vision tasks, and directly transferring them to the PointNet for 3D results in poor performance. We would like to bring to readers’ attention that we have evaluated many popular binarization methods.
> >
> > First, we prove in theory (Theorem 1) that directly applying existing binarization methods on architectures for 3D point clouds results in poor performance as none of them handles feature homogeneity caused by global aggregation functions. Hence, the binarization-friendly EMA we proposed is critical to the success of binary networks on 3D point clouds.
> >
> > Second, we implement as many as five binarization methods on three different architectures in extensive experiments (Table 2 and 3). The experimental results further validate our theory.
> >
> > Moreover, we follow the reviewer’s recommendation and reproduce the XNOR-Net++ (case 1 version with layerwise learnable factors) based on PointNet, our method outperforms XNOR-Net++ by convincing margins. Please refer to A2 for more details.
> >
> > **Q5**: The last layer of ResNet before the linear classifier is a global pooling operation, how does EMA change the results when applied to image classification on a ResNet18 on ImageNet?
> >
> > **A5**: As we check the literature and publicly available codes, the last global pooling layer in ResNet is global average pooling by convention, including the official implementation of ResNet [1] and existing binarized models [2,3]. As we discuss in Sec 3.2.2, average pooling is inherently an EMA. Along with our explanation in A3(2), it is concluded that aggregation function does not impose significant performance degradation in 2D vision on mainstream backbones.
> >
> > Nevertheless, we highlight that this work focuses on point clouds applications, which suffer significantly from model binarization as max pooling is a common design choice for global feature aggregation.
> >
> > [1] He et. al., Deep residual learning for image recognition, CVPR 2016
> >
> > [2] Matthieu, et al. Binarized neural networks: Training deep neural networks with weights and activations constrained to+ 1 or-1, NIPS 2016
> >
> > [3] Liu et. al., Bi-real net: Enhancing the performance of 1-bit cnns with improved representational capability and advanced training algorithm, ECCV 2018

---

> > > ### Author Response · Authors · 2020-11-15
> > > **Response to AnonReviewer #1 (3/3)**
> > >
> > > **Q6**: The cited works don’t support the author statement.
> > >
> > > **A6**: The work cited here are the related 2D vision tasks and the mainstream backbones; we have already cited works for model binarization in the first paragraph of Introduction.  We will modify the introduction in the revised version to avoid confusion.

---

> ### Comment · AnonReviewer1 · 2020-11-20
> **After reading authors' rebuttal**
>
> I appreciate the author’s feedback. The following concerns still remain:
> 1. XNOR-Net++ learns channel-wise scale factors. It seems to me that the proposed LSR learns a  layer-wise scale factor with the initialisation of (10). Could the authors please clarify? If this is the case, to me this is not a significant contribution.
> 2. Moreover, how did the authors initialise XNOR-Net++? For example, it makes sense to use the analytic calculation for this purpose. Furthermore, what if you use (10) for the same purpose? I would be surprised if with proper initialisation a layer-wise learnable scale factor outperforms learnable channel-wise scale factors.
> 3. I’m not convinced that scale distortion is a different problem to the quantisation one encountered in BNNs. I think these problems are essentially the same.
> 4. The paper develops a number of theories to explain, in my opinion, well-known problems in BNNs. Perhaps these problems are encountered in slightly different form because the method operates on point clouts whereas BNNs on images. I’m not convinced that these theories are actually needed to add technical depth to the paper.
> 5. The rebuttal also confirmed (in A5) that one of the paper’s contribution is average pooling. The other contribution of section 3.2  is (as far as I understood) that max pooling on its own does not work, so a shift needs to be applied. This is useful, no question about it, however in my opinion not sufficient. Overall, taking also into account points 1 and 4 above, I still believe that the paper makes a good but not sufficient contribution.

---

> > ### Author Response · Authors · 2020-11-24
> > **Re: “After reading authors' rebuttal” (1/2)**
> >
> > We thank the reviewer’s feedback.
> >
> > We first have to highlight that it is widely accepted that images and point clouds are fundamentally different. As a result, the 3D backbones (such as PointNet) proposed to tackle problems unique to point clouds, are vastly different from classic CNNs for image tasks. Our work is the first to binarize the PointNet for 3D point clouds: we show that the existing binarization methods designed for CNNs cannot be directly applied to PointNet due to the two major challenges (aggregation-induced feature homogenization and scale distortion), to which we develop efficient and easy-to-implement solutions (EMA and LSR).
> >
> > Then we respond to the concerns one by one:
> >
> > **Q1:** XNOR-Net++ learns channel-wise scale factors. LSR learns a layer-wise scale factor with the initialisation of (10). This is not a significant contribution.
> >
> > **A1:** First of all, we have clearly stated in R1A4 of our 15 Nov rebuttal that we implement the Case 1 version of XNOR-Net++ with a layerwise learnable scaling factor for a fair comparison. Please refer to A2 for details.
> >
> > We have already explained in R1A1 of our 15 Nov rebuttal that in Sec 3.3, Theorem 2 shows the binarization causes serious scale distortion and further analyses indicate such scale distortion harms the functionality of the scale-sensitive structures in PointNet. Hence, the initialization of our LSR is designed such that the overall scale of the output feature is recovered to match the full-precision features. In contrast, XNOR-Net++ (Case 1) initializes its scaling factors to minimize quantization errors of parameters in a layer, which does not guarantee proper recovery of output feature scales. As the experiments clearly show that LSR outperforms XNOR-Net++ (Case 1) with layerwise learnable scaling factors by 8.0% in overall accuracy. Compared to XNOR-Net with channelwise scaling factors, LSR still achieves a 4.5% lead. Note that our EMA-max is applied in all experiments. Hence, our theories, supplemented by empirical experiments, have proved that our LSR directly tackles the scale distortion problem of binarized PointNet for 3D point clouds, with its performance far exceeding that of the existing binarization methods.
> >
> > Moreover, we emphasize again that our EMA is essential for all existing binarization methods, including XNOR-Net++, to obtain decent results; when these methods are directly applied to PointNet without EMA, the binarized network achieves extremely low accuracy or even fails to converge, *e.g.*, the accuracy of XNOR-Net++ with the original MAX aggregation function is as low as 4.1%; with our EMA-max applied, however, the performance received a huge improvement of 74.3%.
> >
> >
> > **Q2:** How did the authors initialise XNOR-Net++? For example, it makes sense to use the analytic calculation for this purpose. Furthermore, what if you use (10) for the same purpose? I would be surprised if with proper initialisation a layer-wise learnable scale factor outperforms learnable channel-wise scale factors.
> >
> > **A2:** We need to clarify that Eqn. 10 actually describes the forward and backward propagation of LSR; the initialization is depicted in Eqn. 9.
> >
> > As we mentioned in A1, we have clearly stated in R1A4 of our 15 Nov rebuttal that we implement the Case 1 version of XNOR-Net++ with layerwise learnable scaling factor for a fair comparison (please refer to the original paper for the four versions with different scaling factor settings). Hence, XNOR-Net++ (Case 1) has the same amount of learnable factors as ours.
> >
> > Moreover, we have already used the analytically calculated initialization as the reviewer requested. In fact, XNOR-Net++ does not provide any instruction for initialization in the original paper or release any code; we try our best to implement multiple versions of initialization for XNOR-Net++ and we have already shown the best results we can achieve in the experiments.
> >
> > The experiment results further validate our stance: LSR performs better than XNOR-Net++ with the same amount of parameters (up to 8.0% increase), showing that our LSR, designed to tackle scale distortion, is more effective in improving the accuracy of the binarized PointNet.
> >
> > Besides, applying our initialization in XNOR-Net++ is unreasonable. Our initialization is critical to tackling the scale distortion, and thus an inalienable part of our novel LSR. Applying both EMA and our initialization to XNOR-Net++ essentially makes XNOR-Net++ very similar to BiPointNet. We find this comparison meaningless and the motivation questionable.

---

> > > ### Author Response · Authors · 2020-11-24
> > > **Re: “After reading authors' rebuttal” (2/2)**
> > >
> > > **Q3:** I’m not convinced that scale distortion is a different problem to the quantisation one encountered in BNNs. I think these problems are essentially the same.
> > >
> > > **A3:** We have already explained clearly in our previous response (R1A2 of our 15 Nov rebuttal) and our paper (Sec 3.3) that scale distortion is unique to PointNet for learning on 3D point clouds. A prominent example is the T-Net for feature transformation is sensitive to disturbances in the input scale.
> > >
> > > Specifically, mainstream binarization methods on images focus on the quantization errors between the binarized parameters (weight or/and activation) and the full-precision counterpart, neglecting the impact of binarization on the overall scale of the output features (e.g. IR-Net). Hence, although these works perform reparameterization before quantization, they do not address the scale distortion problem in PointNet for point clouds (our BiPointNet outperforms IR-Net by 22.9%).
> > >
> > > Our BiPointNet outperforms XNOR-Net++ (8.0% increase), XNOR-Net (4.5% increase), and Bi-Real (8.9% increase), which apply the layerwise learnable scaling factors, the channelwise scaling factors for weight and activation, and the channelwise scaling factors just for weight, respectively. The experiments provide strong support to our hypothesis as we are able to outperform the existing binarization methods (with various scaling factors designed to minimize quantization errors) with only one scaling parameter per layer.
> > >
> > >
> > > **Q4:** The paper develops a number of theories to explain, in my opinion, well-known problems in BNNs. Perhaps these problems are encountered in slightly different form because the method operates on point clouts whereas BNNs on images. I’m not convinced that these theories are actually needed to add technical depth to the paper.
> > >
> > > **A4:** We reiterate that PointNet is fundamentally different from 2D CNNs, and we have shown in the paper that 2D binarization methods are NOT readily transferable to 3D (Sec 3). Therefore, what we studied are new problems for the binarization of 3D point cloud networks, and cannot be solved by existing methods.
> > >
> > > For our EMA, **none** of the existing binarization methods are able to address the feature homogenization problem caused by global max pooling with a large kernel size, which is unique to feature learning of PointNet on 3D point clouds. In fact, the binarization on networks for 2D vision hardly encounters similar problems, since the max pooling kernels used in the binarized layers of popular backbones such as ResNet or VGG are small (typically 2x2 or 3x3). However, in networks for 3D point clouds, especially PointNet, max pooling with a large kernel size (even 1024) is often used as the aggregation function.
> > >
> > > Our Theorem 1 proves that the serious homogenization problem is caused by the large kernel size of the aggregate function in binarized PointNet, and our experiments show that without EMA, applying existing binarization methods lead to immense performance drop or even divergence. Most existing binarization methods even cannot achieve 10% accuracy with the original MAX aggregation function, including BNN (7.1%), XNOR-Net++ (4.1%), IR-Net (7.3), Bi-Real (4.0%), and ABC-Net (4.1%). However, the same methods are able to obtain decent results when they are equipped with EMA.
> > >
> > > For our LSR, as we have repeatedly mentioned in A1, A2 and A3, the scale distortion problem is for binarization of PointNet on 3D point clouds, especially for the scale-sensitive structure (such as T-Net), and cannot be solved by existing binarization methods. Our experiments strongly support this view: in A3, the existing binarization methods with various scaling factors for minimizing quantization errors cannot address this problem as well as our BiPointNet. Our BiPointNet is still 4.5% higher than XNOR-Net, while XNOR-Net performs best on binarized PointNet among the existing binarization methods with scaling factors.
> > >
> > > Therefore, both our theories and experiments prove that our BiPointNet caters to 3D point clouds better than the previous binarization work in 2D vision.
> > >
> > >
> > > **Q5:** The other contribution of section 3.2 is (as far as I understood) that max pooling on its own does not work, so a shift needs to be applied. This is useful, no question about it, however in my opinion not sufficient.
> > >
> > > **A5:** Saying the contribution of section 3.2 is a shift for max-pooling is a complete understatement. We would like to stress that EMA-max and EMA-avg are two realizations of EMA, but EMA is not limited to these two variants we develop in the paper. We have stated clearly in both our paper and our previous response that EMA provides a theoretical foundation for designing binarization-friendly aggregation functions on point clouds. EMA-max is only an example that validates our theory. Moreover, we show in Sec 3.2 that EMA explains why the feature homogenization problem is unique to point clouds.

---

### Author Response · Authors · 2020-11-24
**General Response**

We are grateful for the reviewers' positive feedback towards BiPointNet. To assist a clearer understanding of our paper, we summarize our main contributions below:

We present BiPointNet, the first model binarization approach for efficient deep learning on 3D point clouds, to alleviate the resource constraint for real-time point cloud applications that run on edge devices. In this paper, we study the binarization of PointNet for 3D point clouds, and identify the problems with existing binarization methods for 2D vision: aggregation-induced feature homogenization and scale distortion. To solve these problems, we present the Entropy-Maximizing Aggregation (EMA) and Layer-wise Scale Recovery (LSR). We highlight that our techniques are generic, guaranteeing significant improvements on various fundamental tasks and mainstream backbones. Moreover, our BiPointNet is efficient and easy to implement on edge devices, letting it enjoy extremely fast inference in practice while improving the accuracy. BiPointNet gives an impressive $14.7\times$ speedup and $18.9\times$ storage saving on real-world resource-constrained devices, and demonstrates the great potential of binarization.

We also update our manuscripts; the change we made includes:

* In Section 4.2, we add more results about binarization methods (ABC-Net and XNOR-Net++) on PointNet to Table 2.
* In Section 4.2, we add more results about mainstream backbones (PointConv) to Table 3.
* We add more discussions and references about point cloud networks, including PointConv, KPConv, RS-CNN, and ShellNet; we also add discussions and references about mixed-precision quantization.
* In Section 4.3, we report the speed of various binarization methods on the ARM device and present the speed vs accuracy trade-off scatter plot (Figure 5(c)).
* We improve Figure 1 and add related explanations to clarify how LSR works in the whole framework.
* We carefully correct our references in the revised version.

For the detailed explanation, please see our responses to each reviewer.

---

### Decision · Program_Chairs · 2021-01-07
**Final Decision**

**Decision:**

Accept (Poster)

**Comment:**

The authors propose techniques to deal with binarization of 3D point clouds and propose EMA and layer wise scale recovery that improve results across the board for PointNet style models.
An accept.